# Biological Activity and Structural Diversity of Steroids Containing Aromatic Rings, Phosphate Groups, or Halogen Atoms

**DOI:** 10.3390/molecules28145549

**Published:** 2023-07-20

**Authors:** Valery M. Dembitsky

**Affiliations:** Centre for Applied Research, Innovation and Entrepreneurship, Lethbridge College, 3000 College Drive South, Lethbridge, AB T1K 1L6, Canada; valery.dembitsky@lethbridgecollege.ca or dvmioch@gmail.com

**Keywords:** steroids, triterpenoids, isoprenoid lipids, anti-neoplastic, anti-inflammatory, anti-fungal, anti-bacterial, anti-viral, fungal endophytes, plants, marine invertebrates

## Abstract

This review delves into the investigation of the biological activity and structural diversity of steroids and related isoprenoid lipids. The study encompasses various natural compounds, such as steroids with aromatic ring(s), steroid phosphate esters derived from marine invertebrates, and steroids incorporating halogen atoms (I, Br, or Cl). These compounds are either produced by fungi or fungal endophytes or found in extracts of plants, algae, or marine invertebrates. To assess the biological activity of these natural compounds, an extensive examination of referenced literature sources was conducted. The evaluation encompassed in vivo and in vitro studies, as well as the utilization of the QSAR method. Numerous compounds exhibited notable properties such as strong anti-inflammatory, anti-neoplastic, anti-proliferative, anti-hypercholesterolemic, anti-Parkinsonian, diuretic, anti-eczematic, anti-psoriatic, and various other activities. Throughout the review, 3D graphs illustrating the activity of individual steroids are presented alongside images of selected terrestrial or marine organisms. Additionally, the review provides explanations for specific types of biological activity associated with these compounds. The data presented in this review hold scientific interest for academic science as well as practical implications in the fields of pharmacology and practical medicine. The analysis of the biological activity and structural diversity of steroids and related isoprenoid lipids provides valuable insights that can contribute to advancements in both theoretical understanding and applied research.

## 1. Introduction

Natural steroids belong to the class of isoprenoid lipids [1,2]. These metabolites, which can originate from animals, fungi, and plants, exhibit high biological activity and contain a sterane skeleton composed of isoprenoid precursors [3,4,5,6]. Steroids are characterized by the presence of a fused tetracyclic system, such as androstane (**1A**) and related structures, estrane (**1B**), gonane (**1C**), cholestane (**2**), and protostane (**3**) (refer to Figure 1 for their structures) [7,8]. The androstane, cholestane, and/or protostane cores in steroids or triterpenoids can be saturated or partially unsaturated and may incorporate alkyl, hydroxyl, carbonyl, or carboxyl groups [7,8,9]. Isoprenoid lipids, on the other hand, are natural metabolites derived from isoprene molecules and serve various physiological functions while exhibiting a wide range of biological activities [1,2,3,4,5,6].

Protostane-type triterpenoids, predominantly found in plants of the genus *Alisma*, exhibit diverse carbon skeletons and intriguing biological activities [10]. Furthermore, marine- and plant-derived steroids can incorporate various halogens, including chlorine, bromine, or iodine [11,12,13,14]. Notably, seaweeds possess significant nutritional value and have been integral to the diets of many cultures throughout history (depicted in Figure 2). Seaweed extracts are notably abundant in natural growth hormones, known as phytosterols, as well as essential nutrients and trace elements. Algal-derived sterols contribute substantially as the principal lipid component of plant cell membranes and display a broad spectrum of biological activities [15,16,17,18,19,20].

This review provides an overview of the biological activities of steroids and isoprenoid lipids derived from diverse natural sources. Given the extensive number of natural steroids and isoprenoid lipids, we have focused on compounds with established biological activities through experimental studies and computational analyses. This selection aims to cater to pharmacologists, chemists, and researchers from various disciplines who utilize steroids for medicinal purposes.

## 2. Steroids Bearing Aromatic Ring(s)

Steroids bearing aromatic rings are a distinct subgroup within the larger family of steroids, which are characterized by their fused ring structure [21,22,23]. The presence of one or more aromatic rings in these steroids imparts unique chemical and biological properties, making them of particular interest in various fields of research, including medicinal chemistry and drug discovery. Steroids bearing aromatic rings represent a fascinating subgroup of steroids that possess distinct chemical and biological characteristics [22,23,24]. Their unique structural features and diverse pharmacological profiles make them promising candidates for drug development and therapeutic applications. Continued research in this field will expand our knowledge of their biological activities and unlock their potential in various areas of medicine and biology. Natural steroids and triterpenoids that contain one or more aromatic rings in their structure are referred to as aromatic steroids. They are a diverse group of lipid molecules synthesized by bacteria, fungi, plants, invertebrates, and animals [21,22,23,24,25,26]. These aromatic steroids have been identified in various sources, including geological samples, marine sediments, and oil [27,28,29,30,31].

A comprehensive analysis of the literature reveals that the most prevalent subgroup among natural lipids is mono-aromatic steroids and triterpenoids, with an aromatic ring in either position A (approximately 200 metabolites) or position B (around 20 steroids) [32]. Additionally, a small number of di-aromatic steroids have been identified in living organisms, geological samples, marine sediments, and oil, while only a few tri-aromatic steroid hydrocarbons have been found in living organisms, marine sediments, and oil [28,29,30,33,34].

### 2.1. Steroids Bearing Aromatic Ring A in Plants

Steroids bearing an aromatic ring in position A (aromatic ring A) are commonly found in plants, and this contributes to their diverse biological activities. These aromatic steroids play important roles in plant growth, development, and defense mechanisms. Here, we explore the occurrence and functions of steroids with aromatic ring A in plants.

Estrone (**4**, or estra-1,3,5(10)-triene-3-ol-17-one), estradiol (**5**), estriol (**6**), equilin (**7**), hippulin (**8**), and their derivatives (**9**, **10**, **11**, and **12**) represent the well-known examples of mono-aromatic steroids (refer to Figure 3 for their structures). Table 1 provides an overview of their biological activities. Estrone, a female sex hormone, was initially discovered in the 1920s by independent groups of scientists from the USA and Germany [35,36,37,38,39].

Female sex hormonal steroids, specifically estrogens (**4**–**10**), were initially discovered in plants in 1926 by Dohrn and colleagues [40]. Subsequently, other researchers also identified these compounds [41,42,43]. It is noteworthy that hormones such as 17β-estradiol, androsterone, testosterone, and progesterone were found in approximately 80% of the plant species investigated [41]. Estrone (**4**) has been isolated from various plant sources, including the seeds and pollen of *Glossostemon bruguieri*, *Hyphaene thebaica*, *Malus pumila*, *Phoenix dactylifera*, *Punica granatum*, and *Salix caprea*. A sample plant (*Glossostemon bruguieri*) is depicted in Figure 4. Additionally, 17β-estradiol (**8**) was found in the seeds of *Phaseolus vulgaris*, along with estrone (**4**). The distribution of biological activity, exemplified by estrone, is shown in Figure 5. Furthermore, estriol (**6**) has been identified in *Glycyrrhiza glabra* and *Salix* sp. [41,42,43].

Various plant species, including *Brassica campestris*, *Ginkgo biloba*, *Lilium davidii*, and *Zea mays*, have been found to contain total estrogens (**4**–**7**) and 17β-estradiol (**8**) in their pollen and style [44]. Additionally, testosterone has been detected in the pollen of *Pinus bungeana*, *Ginkgo biloba*, and *P. tabulaeformis* [45]. Furthermore, holaromine (**13**), a steroidal alkaloid, has been isolated from the ornamental shrub *Holarrhena floribunda* [46]. Figure 6 illustrates a 3D graph showcasing the predicted and calculated activity of estrone (**4**) as an ovulation inhibitor.

Deoxymiroestrol (**14**), a phytoestrogen, has been isolated from the Thai herb *Pueraria mirifica* [47]. Withanolides (**15**, **19**, and **20**), which are steroids, have been found in various parts of different plants [48]. Jaborosalactone-7 was extracted from *Jaborosa leucotricha*, while jaborosalactone-45 was identified in *Jaborosa laciniata* [49]. In the extract of *Fevillea trilobata* seeds, andirobicin B glucoside (**16**) was discovered [50]. Furthermore, 1-methyl-19-nor-25-D-spirosta-1,3,5(10)-trien-11α-ol (**17**) and its acetate (**18**) were found in the rhizome of *Metanarthecium luteoviride* [51]. The predicted biological activity for mono-aromatic steroids isolated from plants is presented in Table 1. Additionally, Figure 7 illustrates a 3D graph depicting the predicted and calculated anti-neoplastic activity of mono-aromatic ring A plant steroids (**16**, **17**, **21**, **23**, and **24**).

Luvigenin (**21**), a steroid, has been detected in the leaves of *Metanarthecium luteoviride* [52], *Yucca gloriosa* [53], and *Allium giganteum* [54]. Additionally, a cancer-fighting steroid called cayaponoside A4 (**22**) was isolated from the roots and bark of the *Tayuya* tree, which can be found in the Amazon rainforest across Bolivia, Brazil, and Peru [55,56,57].

An unusual triterpene dimer, xuxuasin B (**23**), was isolated from the Brazilian medicinal plant *Maytenus chuchuhuasca* [58]. The leaf extracts and root of *Maytenus ilicifolia* also demonstrated anti-cancer activity and contained a steroid called 6-oxotingenol (**24**) [59,60,61]. In an interesting discovery, an aromatic triterpenoid (**25**) was found in the cones of *Taxodium balticum* extract [62], and it has also been identified among terpenoids in Eocene and Miocene conifer fossils [63]. Furthermore, the bark extract of *Terminalia catappa* contained various compounds, including estrone (**4**), estriol (**6**), equilin (**7**), equilin sulfate (**26**), and a steroid (**27**) [64]. 

#### Steroids Bearing A, B, C, or D Aromatic Ring

Steroids can be categorized based on the presence of an aromatic ring in different positions, such as A, B, C, or D rings [1,7,8,9]. The following are some examples of steroids bearing aromatic rings in these positions. Aromatic A-ring steroids: estradiol: a natural estrogen hormone found in both males and females. Testosterone: the primary male sex hormone responsible for male sexual development and function. Aromatic B-ring steroids: progesterone: a female sex hormone involved in the menstrual cycle and pregnancy. Cortisol: a stress hormone involved in regulating metabolism and immune response. Aromatic C-ring steroids: aldosterone: a hormone that regulates electrolyte balance and blood pressure. Prednisone: a synthetic corticosteroid used as an anti-inflammatory and immunosuppressant. Aromatic D-ring steroids: vitamin D: a group of fat-soluble vitamins important for calcium and phosphate absorption. Calcitriol: the active form of vitamin D involved in calcium regulation and bone health. These are just a few examples of steroids with aromatic rings in different positions. Steroids play various roles in the body, including regulating physiological processes, acting as hormones, and serving as building blocks for other molecules [1,2,3,4,5,6,7,8,9,10,11,12,13,14,15,16].

The compound 3-Hydroxy-19-nor-1,3,5(10),22-cholatetraen-24-oic acid (**25**) is classified as a ring A aromatic bile acid and was discovered in an extract of the Australian sponge *Sollasella moretonensis* [65]. It was also found earlier in human intestinal flora, likely produced by bacteria [66]. Another steroid, a 4-hydroxy-6-oxopregnane-3-glycoside (**29**), which possesses an aromatic ring A, was isolated from a Pohnpei sponge called *Cribrochalina olemda*. Figure 8 depicts the 3D graph representing this compound [67]. Moreover, the extract of the marine sponge *Topsentia* sp. contains geodisterol-3-O-sulfite (**30**), which exhibits anti-fungal activity against *Candida albicans* [68]. In addition to these, a compound named 24,26-cyclo-19-norcholesta-1,3,5(10),22-tetraen-3-ol (**31**) was discovered in the Hainan soft coral *Dendronephthya studeri* [69]. Furthermore, an anti-tumor steroid thioester known as parathiosteroid C (**32**) was identified in the 2-propanol extract of another soft coral species, *Paragorgia* sp. [70].

Mono-aromatic B-ring steroids are a rare group of steroids that can be synthesized by various types of fungi or fungal endophytes. They have also been found in marine sediments and oil deposits. One example is the 19-norergostane skeleton with an aromatic B-ring, known as phycomysterols A (**33**) and C (**34**), which are found in the filamentous fungus *Phycomyces blakesleeanus*. Phycomysterol A has shown anti-HIV activity, as demonstrated by activity analysis. Figure 9 illustrates the 3D graph representing phycomysterol A [71]. The lipid extract of the pathogenic fungus *Fusarium roseum*, also known as *Gibberella zeae*, contained (22*E*,24*R*)-1(10→6)-abeoergosta-5,7,9,22-tetraen-3α-ol (**35**) [72].

Asperfloketal B (**36**), featuring a trioxahexaheterocyclic ring system, was isolated from the sponge-associated fungus *Aspergillus flocculosus* 16D-1 [73]. Furthermore, an aromatic B-ring compound called topsentisterol E1 (**37**) was detected in the bioactive fraction of a marine sponge, *Topsentia* sp. (a sample of the sponge is shown in Figure 10) [74]. Another interesting aromatic B-ring steroid called phomarol (**38**) was produced by a cultured fungus, *Phoma* sp., derived from the giant jellyfish *Nemopilema nomurai* [75]. Additionally, an anti-bacterial lanostanoid, 19-nor-lanosta-5(10),6,8,24-tetraene-1α,3β,12β,22S-tetraol (**39**), was produced by an endophytic fungus called *Diaporthe* sp. LG23, which inhabits the leaves of the Chinese medicinal plant *Mahonia fortunei* [76].

Mono-aromatic C- and D-ring steroids form a rare group of compounds that have been discovered in various sources such as vegetable oils, marine sediments, and petroleum. In the Alberta oil sands, the C20 C-ring mono-aromatic hydroxy steroid acids (**40** and **41**) were found, and it was observed that these compounds can also be synthesized by soil fungi [77]. Steroidal hydrocarbons (**42** and **46**) have been detected in sediments and petroleum samples [78]. An unprecedented sesterterpenoid called phorone A (**43**), featuring an aromatic D ring, was identified in extracts of the Korean sea sponge *Phorbas* sp. [79]. Furthermore, the anti-cancer compound nakiterpiosinone (**44**), which is a C-nor-D homosteroid, was isolated from the sponge *Terpios hoshinota* [80]. Additionally, an intriguing compound called akaol A (**45**), classified as a sesquiterpene quinol, was associated with marine sponges of the genus Aka. The structure of akaol A is depicted in Figure 11 [81].

The extract of *Salpichroa origanifolia* plants, harvested in the provinces of Buenos Aires and Cordoba in Argentina, was found to contain two minor steroids with an aromatic E ring (**47** and **48**) [82]. From the marine sponge *Haliclona* sp., two compounds were identified: terpene-ketide haliclotriol A (**49**) and halicloic acid B (**50**) [83,84]. Steroidal hydrocarbons (**51** and **52**) were isolated from marine sediments and petroleum sources [85,86,87]. Table 2 displays the predicted biological activity for mono-aromatic steroids isolated from various sources, including plants, fungi, invertebrates, marine sediments, and oils. This table provides insights into the potential biological effects and activities associated with these mono-aromatic steroids.

### 2.2. Steroids Bearing Two or Three Aromatic Rings Derived from Natural Sources

Steroids bearing two or three aromatic rings derived from natural sources can be found in various organisms and have diverse biological activities. These are just a few examples of steroids bearing two or three rings that are derived from natural sources. Steroids with complex ring systems can be found in a wide range of organisms and play important roles in biological processes [1,9,78].

Di- and tri-aromatic steroids (**53**–**83**, structures see in Figure 12) represent a small group of natural lipids. These compounds have been isolated and identified in various sources such as marine sediments, oils, and sedimentary rocks [78,85,88]. It is worth noting that di-aromatic steroids, which contain a naphthalene ring, are primarily synthesized by fungi or fungal endophytes [89]. These unique steroids with di-aromatic or tri-aromatic structures contribute to the diversity of natural lipids and their distribution in different environments. Their presence in marine sediments, oils, and sedimentary rocks suggests their relevance in geological and ecological contexts.

In 1936, Canadian biochemist Desmond Beall isolated 6,8-Didehydroestrone (**53**) from the urine of pregnant mares [90]. Additionally, another steroidal hormone called equilenin, specifically estra-1,3,5(10),6,8-pentaen-3-ol-17-one, was also discovered in the urine of pregnant mares in the same year. Subsequently, in 1938, equilenin sulfate (**54**) was isolated from the urine of pregnant mares by Schachter and Marrian [91]. In 1939, it was further synthesized by Bachmann et al. [92]. Moreover, derivatives of equilenin, including 17α-Dihydroequilenin (**55**) and estra-1,3,5,7,9-pentaen-17-one (**56**), were found to be excreted in the urine of horses [93]. These compounds contribute to the understanding of hormonal compositions and metabolic pathways in horses.

The distribution and biological activity of mono-, di-, and tri-aromatic steroids in nature are well-documented. These aromatic steroids are produced by various sources, including microorganisms, fungi, marine invertebrates, plants, animals, marine sediments, and karst deposits. These compounds have demonstrated significant biological activities, including anti-tumor, anti-inflammatory, and neuroprotective effects. The reliability of these activities ranges from 78% to 92%, indicating a high level of confidence in their observed effects. The wide occurrence of aromatic steroids across different natural sources highlights their importance and potential therapeutic applications. Further research and exploration of these compounds could lead to the discovery of novel drugs and therapeutic interventions.

Rare naphthalene-containing steroids (**56**–**59**) have been discovered in the bark of the *Terminalia catappa* tree. It is believed that these naphthalene steroids are synthesized by fungal endophytes that are associated with these plants [89]. Extensive studies of these plants have revealed a wide variety of fungal endophytes present, including species such as *Cercospora* spp., *Cercospora olivascens*, *Colletotrichum gloeosporioides*, *Diaporthe* sp., *Fusarium* sp., *Lasiodiplodia theobromae*, *Pestalotiopsis* spp., *Penicillium* sp., *Penicillium chermesinum*, *Xylaria* sp., *Phoma microchlamidospora*, and *Phomopsis* sp. [94,95]. In addition, a rare di-aromatic steroid (**60**) that contains an unusual naphthyl A/B ring system, resembling equilenin, was isolated from a Hawaiian sponge belonging to the genus *Strongylophora* [96]. Furthermore, a di-aromatic steroid known as (17β,20*R*,22*E*,24*R*)-19-norergosta-1,3,5,7,9,14,22-heptaene (**62**) is produced by the ascomycete fungus *Daldinia concentrica* [97]. These compounds contribute to the diversity of rare di-aromatic steroids and highlight their presence in unique natural sources. 

A diverse range of naphthalene steroid hydrocarbons (**63**–**68**) have been discovered in various natural sources, including marine sediments, fossil plants and algae, ancient fossils, and petroleum [78,98,99,100,101]. These compounds contribute to the wide array of naphthalene-based steroids found in different geological and biological contexts. In contrast, tri-aromatic steroids, or phenanthrene-containing steroids (**69**–**73**) are relatively rare in nature and are found in only a limited number of specimens. One intriguing example is the phenanthrene-containing steroid called cinanthrenol A, which was identified in the marine sponge *Cinachyrella* sp. (a sample of the sponge is depicted in Figure 13). Cinanthrenol A has demonstrated cytotoxic activity against P-388 and HeLa cells and has also shown inhibitory effects on estrogen receptors [102]. These unique phenanthrene-containing steroids exemplify the fascinating diversity of naturally occurring compounds and their potential for various biological activities. Further exploration of these compounds could lead to the discovery of novel therapeutic agents or insights into biological processes.

Acute neurological disorders refer to a group of sudden-onset conditions that affect the nervous system, including the brain, spinal cord, and peripheral nerves. These disorders can arise due to various factors such as infections, trauma, vascular events, metabolic imbalances, autoimmune reactions, or toxic exposures. They are characterized by rapid onset and can lead to severe neurological symptoms and impairments. Figure 14, a 3D graph, illustrates the predicted and calculated activity of an aromatic steroid (**81**) as a potential treatment for acute neurological disorders. The graph demonstrates the relationship between the activity of the compound and its efficacy in treating these disorders. The predicted and calculated activity values, shown on the axes of the graph, represent the potency or effectiveness of the compound in addressing the neurological symptoms associated with acute disorders. The graph also mentions a confidence level of over 92%. This indicates a high degree of certainty in the accuracy of the predicted and calculated activity values. Such confidence levels are typically derived from statistical analysis or predictive modeling techniques used in drug discovery and development. It is important to note that without additional context or information about the specific compound (aromatic steroid **81**), its mechanism of action, and the specific acute neurological disorders being targeted, it is difficult to provide a detailed interpretation of the graph. Further research, clinical trials, and scientific investigation would be necessary to validate the efficacy and safety of the compound as a potential treatment for acute neurological disorders.

Tri-aromatic and/or polyaromatic steroid hydrocarbons are a class of organic compounds that contain three or more aromatic rings fused together with a steroid structure [1,9,78]. These compounds have been identified in various natural sources, including lipid extracts of fossil plants and algae, marine sediments, and petroleum. The presence of tri-aromatic and polyaromatic steroid hydrocarbons in these sources suggests that they have a natural origin and may be formed through the diagenesis and maturation processes of organic matter over time. These compounds often exhibit complex and diverse chemical structures due to the multiple aromatic rings and steroid backbone. The identification and characterization of these compounds have been facilitated by analytical techniques such as gas chromatography-mass spectrometry (GC-MS) and nuclear magnetic resonance (NMR) spectroscopy. Their presence in various geological and biological samples suggests that they may have ecological, physiological, or pharmacological relevance. Investigating their biological activities and potential applications can contribute to our understanding of their functions in nature and may uncover new possibilities for their utilization in various fields.

Tri-aromatic and/or polyaromatic steroid hydrocarbons (**70**–**83**) have been detected in lipid extracts obtained from various sources such as fossil plants, algae, marine sediments, and petroleum [78,85,98,99,103,104]. Among these compounds, an oleanane-related triterpenoid (**80**) with a unique C-2 oxygenated functionality has been identified as the most abundant triterpenoid in a 4900-year-old oak wood sample that was buried in freshwater sediment [105,106]. In addition, other triterpenoids containing phenanthrene structures (**79**, **81**, and **82**) have been found, along with stigmast-4-ene, stigmast-5-ene, stigmastanol, stigmastanol-3-one, 24-ethylcholesta-4,6,22-triene, and β-sitosterol, in fossil cones of *Taxodium balticum*. Stigmastanol-3-one has also been identified in *T. dubium* [107]. Table 3 presents the reported biological activities of mono-aromatic steroids that have been isolated from various sources including fungi, invertebrates, marine sediments, and petroleum. This table provides information on the observed biological effects or properties exhibited by these compounds. However, the specific details of the biological activities mentioned in Table 3 are not provided in the given text.

Further research is needed to fully understand the roles and significance of tri-aromatic and polyaromatic steroid hydrocarbons in natural systems. Their presence in various geological and biological samples suggests that they may have ecological, physiological, or pharmacological relevance. Investigating their biological activities and potential applications can contribute to our understanding of their functions in nature and may uncover new possibilities for their utilization in various fields.

## 3. Steroids Bearing Phosphate Esters

Phosphorus, with an atomic number of 15, is a prevalent chemical element found in both the earth’s crust and seawater [108,109,110]. Its discovery dates back approximately 350 years [111]. Due to its high reactivity, phosphorus is typically found in nature in the form of phosphates, which are salts of phosphoric acid [112]. Apatite, a mineral compound, is considered one of the most significant sources of phosphorus [113,114].

Steroids bearing phosphate esters are a class of organic compounds that combine the structure of steroids with phosphate groups attached to specific positions. These phosphate esters can be covalently linked to the steroid molecule, typically through ester bonds. The addition of phosphate esters to steroids introduces new chemical properties and functional groups, which can have significant effects on the compound’s biological activity and physiological functions. Phosphate esters play important roles in cellular signaling, energy metabolism, and various biochemical processes. Phosphate esters in steroids can also serve as important intermediates in metabolic pathways. For instance, in the biosynthesis of steroid hormones, phosphate esters are involved in the conversion of cholesterol to various hormone precursors, such as pregnenolone. Furthermore, some steroid-based drugs utilize phosphate esters to enhance their pharmacological properties. By introducing phosphate groups, these compounds can exhibit improved solubility, bioavailability, and targeted delivery to specific tissues or cells. Overall, steroids bearing phosphate esters are biologically significant molecules that contribute to cellular processes, membrane structure, and the modulation of hormonal activities. Understanding their synthesis, functions, and interactions is crucial in unraveling the complexities of biological systems and developing therapeutic interventions [115,116,117,118,119,120,121].

### Steroid Phosphate Esters in Marine Invertebrates

Steroid phosphates (**84**–**87**), as shown in Figure 15, were first discovered by Italian scientists from the University of Federico II approximately three decades ago. Their discovery came during the study of polar lipids extracted from the deep marine starfish *Tremaster novaecaledoniae* [122]. The isolated glycosides obtained from this research were named tremasterols A–C (**84**, activity is shown in Table 4), along with compounds **85** and **86**. Figure 16 illustrates the distribution of biological activity, specifically for tremasterol (**84**), represented as a percentage. This graph provides insights into the effectiveness or impact of tremasterol in various biological contexts. The identification and characterization of these steroid phosphates from the marine starfish *T. novaecaledoniae* represent significant contributions to the field of natural product research. Further investigations are likely needed to fully understand the biological activities and potential applications of these compounds, including their mechanisms of action and potential therapeutic benefits.

Phosphorylated sterol sulfates, known as haplosamates A (**88**) and B (**90**) and minor secosteroid (**89**), were discovered in a marine sponge species called *Cribrochalina* sp. [123]. Haplosamate A is distinguished by its unique C28 sterol structure, featuring a sulfate group at C-3 and a methyl phosphate at position 15. Haplosamate B, on the other hand, contains two phosphate groups at positions 7 and 15 [123]. The 3D graph illustrating the activity of haplosamate A (**88**) is depicted in Figure 17. Further semi-synthetic analogues, including compounds **91**–**94**, have also been isolated and studied. Desulfohaplosamate (**95**), haplosamate A (**88**), and other steroid analogues (**96**–**99**) were evaluated for their interaction with CB1 and CB2 cannabinoid receptors through binding tests [124]. It is worth noting that both steroids containing a phosphate group, namely **88** and **90**, were discovered in the polar organic fraction of an Indonesian sponge species called *Dasychalina* sp. (shown in Figure 18) [124]. The identification and evaluation of these phosphorylated sterol sulfates and their analogues provide valuable insights into their potential biological activities and interactions. Further research is necessary to fully understand their mechanisms of action, therapeutic potential, and roles within marine ecosystems.

For comparing biological activity, several semi-synthetic steroids have been selected. These include prednisone phosphate (**99**), testosterone 17β-phosphate (**100**), cortisol 21-phosphate (**101**), and cholesterol 3β-phosphate (**102**). Prednisone phosphate (**99**) has been shown to possess anti-inflammatory activity [125]. This property makes it useful in the treatment of various inflammatory conditions. Testosterone 17β-phosphate (**100**) is an androgen and belongs to the class of anabolic steroids. It is commonly used for intramuscular injections and is known for its anabolic effects on muscle growth. Additionally, it serves as a substrate for phosphatases in the phosphatase pool of the prostate [126]. Cortisol 21-phosphate (**101**) is a glucocorticoid that plays a crucial role in regulating various physiological processes. It is involved in the regulation of metabolism, immune responses, and stress responses. The phosphate group attached to cortisol 21 enhances its solubility and may influence its activity. Cholesterol 3β-phosphate (**102**) is a modified form of cholesterol with a phosphate group attached to its 3β position. The addition of the phosphate group introduces new chemical properties to cholesterol, potentially influencing its functions and interactions within the body. These semi-synthetic steroids have been selected for comparison with steroids isolated from marine invertebrates in order to gain insights into their biological activities and potential applications. Further research is necessary to fully understand the specific mechanisms of action and therapeutic implications of these compounds.

Cortisol 21-phosphate (**101**), as depicted in Figure 19, belongs to the glucocorticoid class of hormones. It functions to increase blood sugar levels through gluconeogenesis and promotes the metabolism of fats, proteins, and carbohydrates. Additionally, cortisol 21-phosphate serves as a substrate for alkaline phosphatase and finds utility in enzyme immunoassays for human chorionic gonadotropin, human growth hormone, α-fetoprotein, and estradiol [127]. The activities and properties of cortisol 21-phosphate can be found in Table 5, and its 3D graph is illustrated in Figure 20.

Cholesterol 3β-phosphate (**102**) is known for its role in promoting the normalization of blood pressure and its involvement in atherogenesis, the process of plaque formation in arteries [128,129]. Two cholesterol-lowering agents, sodium ascorbyl campestanol phosphate (**103**) and sodium ascorbyl sitostanol phosphate (**104**), have been derived from cholesterol and extensively studied [130]. Furthermore, two semi-synthetic steroidal phosphate esters (**105** and **106**, 3D graph is illustrated in Figure 21), are identified as inhibitors of cholesterol biosynthesis. These compounds show potential for the treatment or prevention of atherosclerosis, a major contributor to cardiovascular disease [131]. The investigation and understanding of these steroidal phosphate compounds contribute to advancements in the field of hormone research and lipid metabolism and the development of potential therapeutic interventions for various conditions, including atherosclerosis and related cardiovascular disorders.

Compound (**107**) is a steroid phosphate ester that incorporates pivalic acid. This compound, known as the anionic chemical delivery system (ACDS), was specifically developed to facilitate the delivery of testosterone to the brain. By enhancing its lipophilicity, systemically administered T-ACDS can passively traverse the blood–brain barrier. The effectiveness of this tested drug has been demonstrated [132]. Estradiol phosphates (**108** and **109**) are esters of estrogen that are combined with phosphoric acid. These compounds serve as prodrugs of estradiol within the human body. In medical practice, both drugs have been utilized for the treatment of prostate cancer [133]. Betamethasone sodium phosphate (**110**) has been synthesized and is employed in the treatment of various conditions such as asthma, allergies, arthritis, Crohn’s disease, ulcerative colitis, and adrenal disease [134]. The development and utilization of these compounds highlight the ongoing advancements in drug development and therapeutic approaches. However, it is crucial to consult with healthcare professionals for proper guidance and administration of these medications, considering individual patient factors and specific medical conditions.

Several steroid phosphate esters, namely compounds **111** to **114** (3D graph is shiwn in Figure 22), have been identified in the eggs of the desert locust, *Schistocera gregaria*. It is intriguing to note the presence of these steroids in deferred eggs, although their specific origin remains unknown [135]. The detection of these compounds in locust eggs raises interesting questions about their potential roles and functions in the reproductive processes of the species. However, it is important to highlight that the biological activity of these compounds obtained from locust eggs has not been investigated or characterized. 

**Figure 20 molecules-28-05549-f020:**
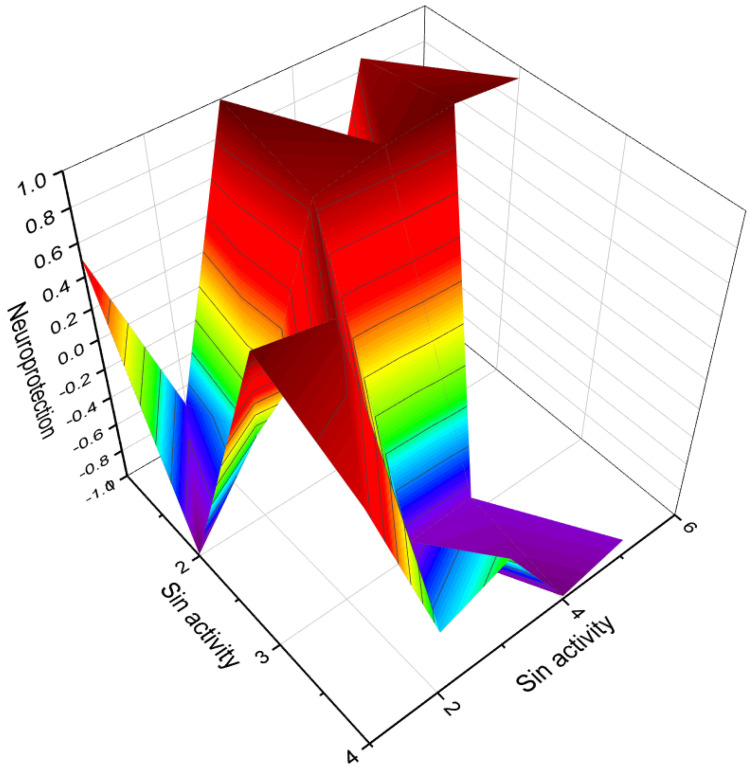
A 3D graph presenting the predicted and calculated activity for neuroprotection of steroid phosphate esters, specifically compounds **100**, **101**, **108**, and **109**. The graph demonstrates the relationship between the activity of these compounds and their efficacy in promoting neuroprotection. The predicted and calculated activity values, shown on the axes of the graph, represent the potency or effectiveness of the steroid phosphate esters in terms of their neuroprotective properties. With a confidence level of over 96%, the graph indicates a high degree of certainty in the accuracy of the predicted and calculated activity values. Neuroprotection is a critical aspect of research and development in the field of neuroscience, aiming to identify compounds that can preserve and protect neurons from damage or degeneration. The evaluation of steroid phosphate esters for their neuroprotective activity provides valuable insights into their potential applications in treating neurological disorders or promoting overall brain health. The concept of neuroprotection has gained significant attention in the field of neuroscience and neurology, particularly in the context of neurodegenerative diseases, stroke, traumatic brain injury, and other conditions that involve neuronal damage. Ayurveda, a centuries-old Indian traditional medicine practice, incorporates the use of herbal extracts and plant-based remedies to address a range of neuropsychiatric disorders [136,137,138]. This ancient healing system recognizes the potential of natural compounds derived from herbs and plants in promoting neurological and mental well-being. In recent times, scientific research has provided evidence supporting the neuroprotective properties of steroid phosphate esters derived from invertebrates or their semi-synthetic analogues. These compounds have demonstrated significant efficacy in safeguarding neurons and mitigating neurodegenerative processes. The exploration of steroid phosphate esters derived from invertebrates, or their synthetic counterparts, as potential neuroprotective agents is an exciting area of research. These compounds hold promise in the development of novel therapeutic interventions for various neurological conditions and disorders.

**Figure 21 molecules-28-05549-f021:**
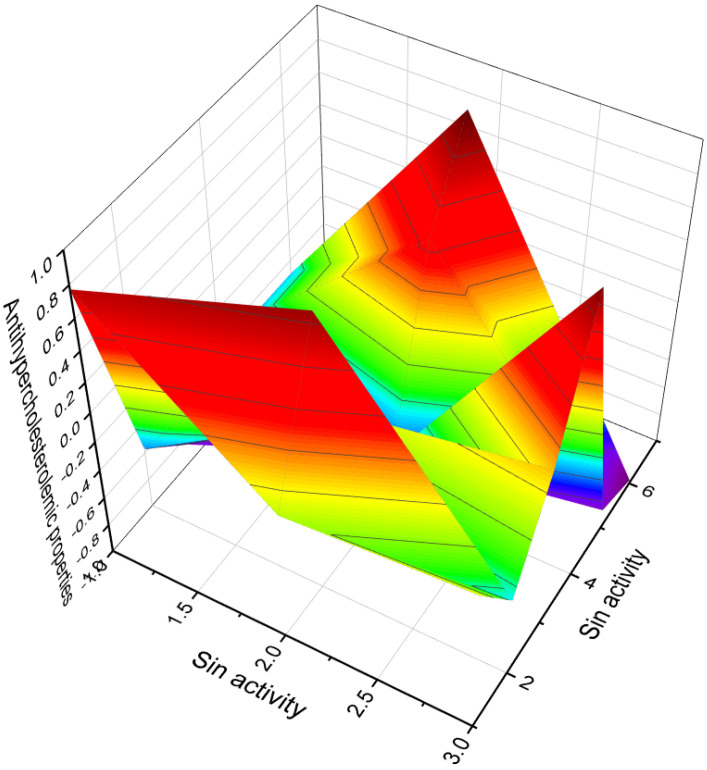
A 3D graph of the predicted and calculated anti-hypercholesterolemic activity of the steroid phosphate esters **104**, **105**, **106**. With a confidence level of over 96%, the graph reflects a high degree of certainty in the accuracy of the predicted and calculated activity values. It is noteworthy that various plants, such as *Hemidesmus indicus*, *Pulicaria gnaphalodes*, *Pandanus tectorius fruits*, *Buchholzia coriacea*, and *Swietenia mahagoni*, have been recognized for their anti-hypercholesterolemic properties, as demonstrated by their extracts [138,139,140,141,142]. Additionally, extracts from brown algae and the cyanobacterium *Arthrospira platensis* have also shown an anti-hypercholesterolemic effect. Interestingly, steroid phosphate esters derived from invertebrates exhibit strong anti-hypercholesterolemic properties. This highlights the potential of these compounds as promising candidates for the development of therapeutic interventions aimed at managing high cholesterol levels.

**Figure 22 molecules-28-05549-f022:**
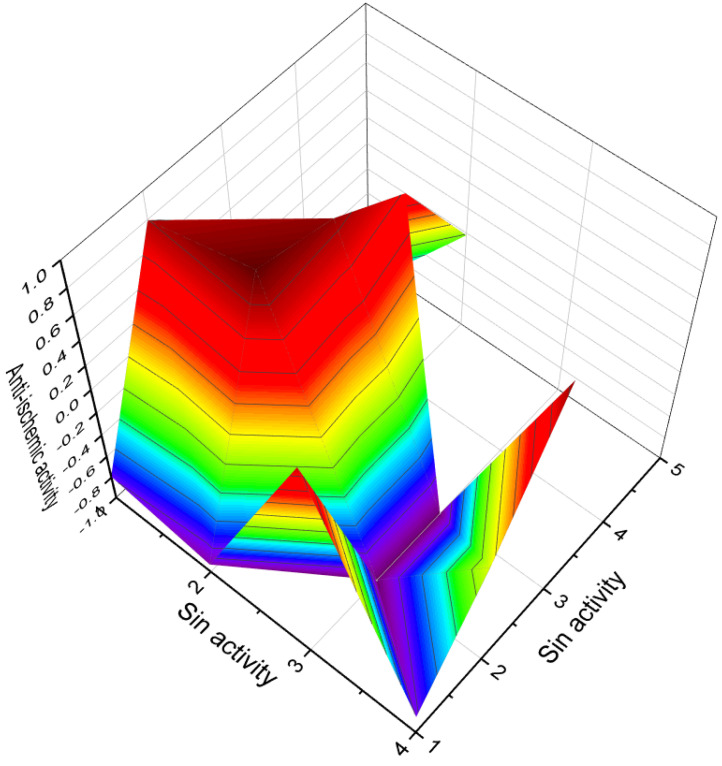
A 3D graph illustrating the predicted and calculated anti-ischemic activity of steroid phosphate esters, specifically compounds **111**, **112**, **113**, and **114**. The graph depicts the relationship between the activity of these compounds and their effectiveness in reducing myocardial oxygen consumption during ischemic conditions. The predicted and calculated activity values, depicted on the axes of the graph, represent the potency or efficacy of the steroid phosphate esters in terms of their anti-ischemic properties. The confidence level of over 93% indicates a high degree of certainty in the accuracy of the predicted and calculated activity values. Anti-ischemic activity refers to the ability of a compound to mitigate the detrimental effects of reduced blood flow and inadequate oxygen supply to the heart muscle. By reducing myocardial oxygen consumption, these steroid phosphate esters hold promise in preventing or alleviating ischemic episodes and related cardiac complications. It is important to note that further research, including experimental validation and clinical trials, is necessary to fully understand the mechanisms of action and optimal applications of these steroid phosphate esters as anti-ischemic agents. Their potential therapeutic implications in the context of ischemic heart disease warrant exploration to develop effective treatments for this condition.

Understanding the presence and activities of steroid phosphate esters in locust eggs may contribute to our knowledge of reproductive biology, insect development, and the hormonal regulation of insect populations. Further research is warranted to explore the biological properties and potential functions of these compounds in the context of locust biology.

## 4. Steroids Bearing a Halogen Atom (Cl, Br, or I)

Natural halogenated steroids are a class of organic compounds that contain halogen atoms (such as fluorine, chlorine, bromine, or iodine) attached to the steroid structure. These compounds can be found in various natural sources, including marine organisms, plants, and microorganisms [143,144,145,146,147].

Halogenated steroids often exhibit unique chemical and biological properties due to the presence of halogen atoms. The incorporation of halogens into the steroid structure can affect the compound’s stability, lipophilicity, and interactions with biological systems. Marine organisms, particularly marine sponges, are known to produce a wide range of halogenated steroids. These compounds are believed to play important roles in the defense mechanisms of these organisms, protecting them against predators and pathogens. Halogenated steroids from marine sources have been the subject of extensive research due to their diverse biological activities and potential therapeutic applications. Some of these compounds have demonstrated anti-microbial, anti-viral, anti-inflammatory, and anti-cancer properties [143,144,145,146,147,148,149,150,151,152,153,154,155,156].

### 4.1. Chlorinated Plant Steroids

Chlorinated plant steroids are a specific subset of plant steroids that contain chlorine atoms attached to their chemical structure. These compounds are derived from plants and exhibit unique properties and potential biological activities due to the presence of chlorine atoms [143,146,148,157]. These chlorinated plant steroids can be found in various plant species, particularly those that have adapted to environments with high chlorine levels, such as salt marshes or coastal areas. These compounds are believed to play a role in the plants’ adaptation to such environments, helping them cope with salinity stress or other ecological factors.

Chlorinated plant steroids have also been investigated for their potential as bioactive compounds with pharmacological applications. However, further research is needed to fully understand their mechanisms of action, physiological functions, and potential therapeutic uses. It is important to note that the presence and biological activities of chlorinated plant steroids can vary among different plant species. Studying these compounds can provide valuable insights into plant adaptations to challenging environments and may contribute to the discovery of novel bioactive compounds with pharmaceutical or agricultural significance. Research on chlorinated plant steroids is still relatively limited compared to other classes of plant steroids. However, some studies have identified and characterized specific chlorinated plant steroids and explored their potential biological activities [145,146,147,148,149,150,151,152,153,154,155,156].

The discovery of chlorine-containing steroids began with the isolation of jaborosalactone C (**115**) and jaborosalactone E (**116**) from the leaves of the *Jaborosa integrifolia* plant, which belongs to the Solanaceae family (a representative plant is shown in Figure 23) [158]. These compounds represent the first identified chlorine-containing steroids. In addition, the *Acnistus breviflorus* plant has been found to produce steroids such as compound **116** and compound **117**, which possess cytostatic activity. Similarly, cytotoxic withanolide (**117**, structure seen in Figure 24) has been isolated from *Withania frutescens*, another plant from the Solanaceae family [159].

Physalolactone C (**118**), displayed in a 3D graph in Figure 25, was identified in the fruits of *Physalis peruviana* (Cape gooseberry) [160]. This compound is structurally similar to the aforementioned steroids and exhibits cytotoxic properties. Additionally, from the same plant, physalolactone (**119**) was obtained from the roots, and a minor steroid of the leaves, 4-deoxyphysalolactone (**120**), was extracted [161].

Physaguline B (**121**, activity shown in Table 6) was discovered in *Physalis angulata* [162]. This compound represents a chlorinated sterol found in the plant, expanding our knowledge of the chemical diversity within *Physalis* species. Withanolide D chlorohydrin (**122**), presented in a 3D graph in Figure 26, was identified in *Withania somnifera*, commonly known as Ashwagandha, while (**119**) and (**123**) were discovered in *Acnistus breviflorus* [163,164]. Further research on *W. somnifera* revealed the presence of withanolide C (**123**), (**119**), and (**124**). These compounds were also found in *Dunalia tubulosa*, which belongs to the Solanaceae family, closely related to the plants [165].

Jaborochlorodiol (**125**) and jaborochlorotriol (**126**), representing a new structural type of chlorinated steroid, were identified in extracts from *Jaborosa magellanica*, a flowering plant of the Solanaceae family found in Punta Arenas, Chile [166]. Furthermore, the aerial parts of *Tolpis proustii* and *T. lagopoda*, native to La Gomera, Canary Islands, led to the isolation of chlorinated sterols: 30-chloro-3β-acetoxy-22α-hydroxyl-20(21)-taraxastene (**127**) and its acetylated analogue (**128**). In vitro antioxidant activities of the extracts were evaluated using the DPPH and ABTS scavenging methods. The cytotoxicity of isolated compounds demonstrated activity against the human myeloid leukemia K-562 and K-562/ADR cell lines [167].

Withanolide Z (**129**) was isolated from *Withania somnifera* as an inhibitor of topoisomerase I from the parasite *Leishmania donovani*, suggesting its potential in anti-parasitic applications [168]. Cytotoxic phyperunolides C (**130**) were found in the leaves of *Physalis peruviana* [169,170], highlighting their potential cytotoxic properties. Hsieh et al. [171] isolated cytotoxic tubocapsenolide G (**131**) from *Tubocapsicum anomalum*.

Physagulin I (**132**, the 3D graph is shown in Figure 27), a 14β-hydroxywithanolide, has been isolated from *Physalis* species and possesses an α-oxygenated functionality at position 15 [172]. Additionally, jaborosalactol 23 (**133**), another 14β-hydroxywithanolide, has been identified in *Jaborosa bergii*, a flowering plant in the Solanaceae family [173]. Nicotra et al. [174] reported the isomeric chlorohydrin, jaborosalactone 37 (**134**, structure seen in Figure 28, and activity see in Table 7), from *Jaborosa rotacea*, and jaborosalactone T (**135**) was isolated from *Jaborosa sativa* (synonym *Trechonaetes sativa*) collected in Argentina [175]. Anomanolide D (**136**), identified as the 16α-hydroxy substituent, was discovered in the fruits of *Tubocapsicum anomalum* collected in Japan [176]. Additionally, tubonolide A (**137**, the 3D graph is shown in Figure 29), a 16,17-dihydroxylated withajardin, was found in the same plant [177].

Unusual 15,21-cyclowithanolides of the norbornane type, jaborosalactols 21 (**138**) and 22 (**139**), were isolated from *Jaborosa bergii* [178]. Furthermore, the acid hydrolysate of a methanolic extract of *Tubocapsicum anomalum* contained TH-6 (**140**) [179]. These discoveries highlight the occurrence of chlorine-containing steroids in plants, particularly in the Solanaceae family. The identification and characterization of these compounds contribute to our understanding of the chemical diversity of natural products and their potential biological activities. Further research is needed to explore the mechanisms of action and therapeutic applications of these chlorine-containing steroids in various fields, including medicine and agriculture.

A group of spiranoid withanolides with a 17(20)-ene-22-keto system, namely jaborosalactones 3 (**142**) and 6 (**143**), were isolated from *Jaborosa runcinata* collected in Argentina [180]. These compounds represent chlorinated steroids with unique structural features. Additionally, jaborosalactone 10 (**141**), presented in a 3D graph in Figure 30, was found in both *J. runcinata* and *J. odonelliana* [181]. This compound further expands our understanding of the chemical diversity within the *Jaborosa* genus.

Two chlorinated 24,25-epoxy-γ-lactols (**144** and **145**) were isolated from plants of *Jaborosa parviflora* [182]. These compounds possess a chlorine atom and an epoxy group within their structures, contributing to their distinctive properties. Furthermore, the chlorohydrins jaborosalactone 42 (**146**) and jaborosalactone 49 (**147**) were detected in *Jaborosa caulescens* var. *bipinnatifida* [183] and *Jaborosa laciniata* [184]. These compounds exhibit a chlorohydrin moiety, further enhancing the chemical diversity within the *Jaborosa* species.

A group of constituents called physalins, which belong to the 13,14-seco-16,24-cycloergostane class of compounds, have been identified in extracts of *Brachistus stramoniifolius*, *Margaranthus solanaceous* (sub nom. *Physalis solanaceous*), and *Schraderanthus viscosus* (sub nom. *Saracha viscosa*) [185,186,187]. These compounds, including physalins **148**, **149**, **150**, and **151** (the 3D graph is shown in Figure 31), exhibit unique structural characteristics within the 13,14-seco-16,24-cycloergostane framework.

Two withanolides with a hemiketal bridge between what was originally ketone functions at C-12 and C-22 have also been discovered. Upon formation of the D-lactone, these compounds, known as **152** and **153**, were detected and identified from *Jaborosa rotacea* [188]. These compounds demonstrate a distinct structural arrangement, featuring a six-membered ring with a β-oriented hydroxy group at C-12 and a spiroketal at C-22. Figure 28 and Figure 32 showcase the structures of various steroids, providing an overview of the diversity within the class. Furthermore, Table 1 presents the biological activities associated with plant chlorinated steroids, highlighting their cytostatic, anti-neoplastic, anti-eczematic, anti-diabetic, anti-bacterial, and other activities. These chlorinated steroids exhibit a range of characteristic biological activities, indicating their potential significance in various fields, including medicine, pharmacology, and agriculture. However, it is important to conduct further research, including in vitro and in vivo studies, to fully understand the mechanisms of action, therapeutic potential, and safety profile of these compounds.

### 4.2. Halogenated Steroids Derived from Marine Sources

Halogenated steroids derived from marine sources are natural compounds that contain halogen atoms (such as chlorine, bromine, or iodine) and are obtained from various marine organisms. These marine organisms can include algae, sponges, corals, mollusks, and other marine invertebrates. Halogenated steroids from marine sources exhibit diverse chemical structures and biological activities, making them of interest in the fields of pharmacology and drug discovery [143,145,147,154].

Strong cytotoxic chlorinated steroids known as clionastatins A (**154**) and B (**155**) have been discovered in the burrowing sponge *Cliona nigricans*. The structures of marine steroids can be observed in Figure 32 and activity see in Table 8. These remarkable compounds contain tri- and tetrachlorinated androstane derivatives, respectively. They are considered the first polyhalogenated steroids found in a living organism, whether marine or terrestrial, and represent the first instances of halogenated androstanes in nature [189]. Clionastatins A and B exhibit potent cytotoxic activity, making them of significant interest in the field of cancer research and drug development. These compounds have shown the ability to inhibit the growth of cancer cells in vitro and have demonstrated promising anti-cancer potential. 

The discovery of clionastatins A and B highlights the unique chemistry and biodiversity found in marine organisms. These compounds contribute to our understanding of the natural products derived from marine sources and their potential therapeutic applications. Further research is needed to elucidate the precise mechanisms of action and therapeutic potential of clionastatins A and B, as well as to explore their structure–activity relationships. Investigating these compounds can provide insights into the development of novel anti-cancer agents and inspire the discovery of additional halogenated steroids derived from marine organisms.

Aragusterol C (**156**), a chlorinated steroid, was isolated from an Okinawan marine sponge of the genus *Xestospongia* sp. This compound exhibited strong inhibitory effects on the proliferation of KB cells in vitro. Furthermore, it demonstrated potent in vivo anti-tumor activity against L1210 cells in mice [190]. The distribution of biological activity percentages for aragusterol C is depicted in Figure 33. Another compound, aragusteroketal C (**157**), which is a steroid with a dimethylketal structure, was also isolated from the same sponge. This chlorinated steroid displayed cytotoxic activity against the KB tumor cell line, with an IC_50_ value of 4 ng/mL [191]. Additionally, a chlorinated steroid (**158**) was isolated from the soft coral *Sinularia brassica*. This coral-derived compound offers unique structural and chemical characteristics [192]. The coral sample associated with this compound is shown in Figure 34.

Cytotoxic chlorinated ketosteroids known as kiheisterones C (**159**), D (**160**), and E (**161**) were discovered in the extracts of the marine sponge *Strongylacedon* sp. from Maui [193]. These compounds exhibit cytotoxic activity and represent an intriguing class of chlorinated ketosteroids derived from a marine source. In addition, unique pentacyclic saturated sesterpenes condensed with a hydroxy-hydroquinone moiety, known as 6′-chlorodisidein (**162**) and 6′-bromodisidein (**163**), have been isolated from the marine sponge *Disidea pallescens* in the form of disulfate sodium calcium salts [194]. These compounds possess a distinct structural arrangement, incorporating both chlorine and bromine atoms. The discovery of these chlorinated compounds further highlights the chemical diversity and pharmacological potential of natural products derived from marine organisms. The cytotoxic and unique structural characteristics of kiheisterones and disideins offer promising avenues for further exploration in the fields of cancer research and drug development.

Chalinulasterol (**164**), a chlorinated sterol disulfate, was isolated from the Caribbean sponge *Chalinula molitba* [195]. This compound represents a unique chlorinated sterol derivative found in a marine organism. Nakiterpiosinone (**165**) and nakiterpiosin (**166**), two related C-nor-D homosteroids, were identified in MeOH extracts of the sponge *Terpios hoshinota*. These compounds have shown potential as anti-cancer agents, particularly in tumors resistant to existing anti-mitotic agents and dependent on Hedgehog pathway responses for growth [196,197]. Their discovery highlights the importance of exploring marine sources for novel compounds with therapeutic potential.

The marine sponge *Topsentia* sp. yielded a chlorine-containing steroid sulfate (**167**) and the first natural iodinated steroid (**168**) [198]. These compounds showcase the chemical diversity of halogenated steroids derived from marine sources and contribute to our understanding of the unique natural products found in marine organisms. Chlorinated stypotriol triacetate (**169**) was detected in the dichloromethane extract of the brown alga *Stypopodium flabelliforme* [199]. This compound represents a chlorinated derivative of stypotriol, a sterol commonly found in brown algae. The identification of chlorinated derivatives expands our knowledge of the chemical variations within marine sterols. Furthermore, the (3β,5α,22*R*,23*S*)-22-chlorocholesta-8,14-diene-3,23-diol (**170**) was found in MeOH-CHCl_3_ extracts of the starfish *Echinaster sepositus* [200]. This chlorinated steroid exhibits a unique structural arrangement and represents an interesting discovery in the field of marine natural products.

Two unique chloro-pregnane steroids (**171** and **172**) have been isolated from the eastern Pacific octocoral *Carijoa multiflora* [201]. These compounds exhibit distinct structures and represent novel chlorinated steroids found in the marine environment. The 3D graph depicting the predicted and calculated activity for compound **171** is shown in Figure 35. In addition, three chlorinated steroids, namely yonarasterols G (**173**), H (**174**), and I (**175**), were discovered in MeOH extracts of the Okinawan soft coral *Clavularia viridis* [202]. These compounds contribute to the growing repertoire of chlorinated steroids derived from marine sources. These compounds exhibit diverse chemical architectures and display unique halogenation patterns that contribute to their biological activities. The biological activities of marine halogenated steroids are varied, with anti-tumor, anti-fungal, anti-cancer, and anti-bacterial activities being characteristic among the compounds. Particularly, anti-cancer activity appears to be a common feature observed in the presented steroids.

## 5. Conclusions

This comprehensive review has explored the diverse range of biological activity and structural variations found within steroids and related isoprenoid lipids. The analysis encompassed various natural compounds, including steroids with aromatic ring(s), steroid phosphate esters from marine invertebrates, and steroids bearing halogen atoms (I, Br, or Cl). These compounds are derived from sources such as fungi, fungal endophytes, plants, algae, and marine invertebrates. Through an examination of referenced literature sources, their biological activity was evaluated through in vivo and in vitro studies, as well as employing the QSAR method. The findings revealed a multitude of compounds exhibiting remarkable properties, including strong anti-neoplastic, anti-proliferative, anti-hypercholesterolemic, anti-Parkinsonian, anti-eczematic, anti-psoriatic, and various other activities. To enhance comprehension, the review incorporated visual aids such as 3D graphs illustrating the activity of individual steroids and images showcasing selected terrestrial or marine organisms. Furthermore, the review provided explanations elucidating certain types of biological activity associated with these compounds. Overall, the findings presented in this review not only contribute to the academic scientific knowledge in the field but also hold practical relevance for the development of pharmacological interventions and advancements in practical medicine. The review utilized data from various authors regarding the biological activity of natural steroids. To assess the potential activity of these steroids, the PASS program was employed. The PASS program utilizes structural features of compounds to predict their biological activity profiles. By inputting the structural information of the natural steroids into the program, their potential activity across multiple predefined activity classes was estimated. However, it is important to note that these predictions are based solely on structural information and should be validated through experimental studies.

## Figures and Tables

**Figure 1 molecules-28-05549-f001:**
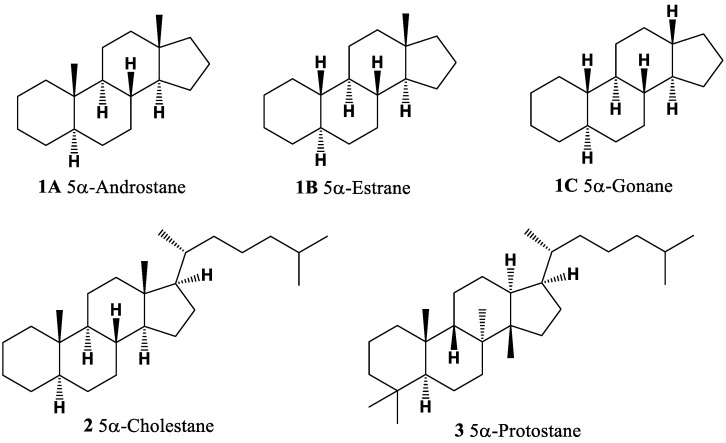
Androstane, cholestane, and protostane are steroid or triterpenoid core structures.

**Figure 2 molecules-28-05549-f002:**
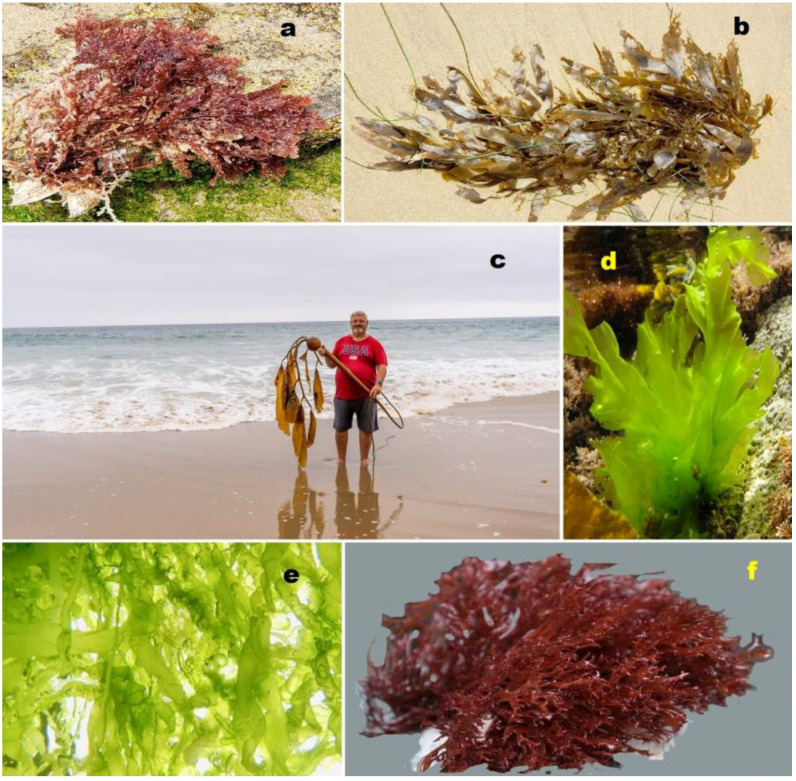
The red and brown algae macrophytes are abundant sources of biologically active metabolites, including steroids. Here, we highlight some representative examples: (**a**) *Laurencia pacifica* (red alga, Rhodophyceae). This species is known for producing halogenated metabolites, such as sesquiterpenes, diterpenes, triterpenes, and C15 acetogenins. (**b**) *Laminaria digitata* (brown alga, Phaeophyceae). A commonly consumed brown algae, particularly in coastal regions, with kelp that is rich in terpenoids, essential amino acids, polyunsaturated fatty acids, carbohydrates, vitamins, and minerals such as iron and calcium. (**c**) *Sargassum* sp. (brown alga, Fucales). Various species of the *Sargassum* genus are utilized for human nutrition and serve as a valuable source of steroids, proteins, vitamins, carotenoids, and minerals, and Professor Dembitsky collected the biological material in Southern California, summer 2018. (**d**) *Ulva lactuca* (sea lettuce, green alga, Ulvaceae). Cultivated in China, Republic of Korea, and Japan, sea lettuce is consumed by manatees, sea slugs, and shellfish. Extracts of this edible green algae contain bioactive components, including steroids and triterpenoids. (**e**) *Enteromorpha intestinalis* (green bait, sea lettuce, green alga, Ulvaceae). This green alga, commonly known as green bait or sea lettuce, produces a wide range of terpenoids, including steroids. (**f**) *Gracilaria pacifica* (red spaghetti, red alga). Widely used in the cosmetic industry for shampoos, creams, soaps, and sunscreens, this red alga contains terpenoids and carotenoids and serves as a source of high-quality agar.

**Figure 3 molecules-28-05549-f003:**
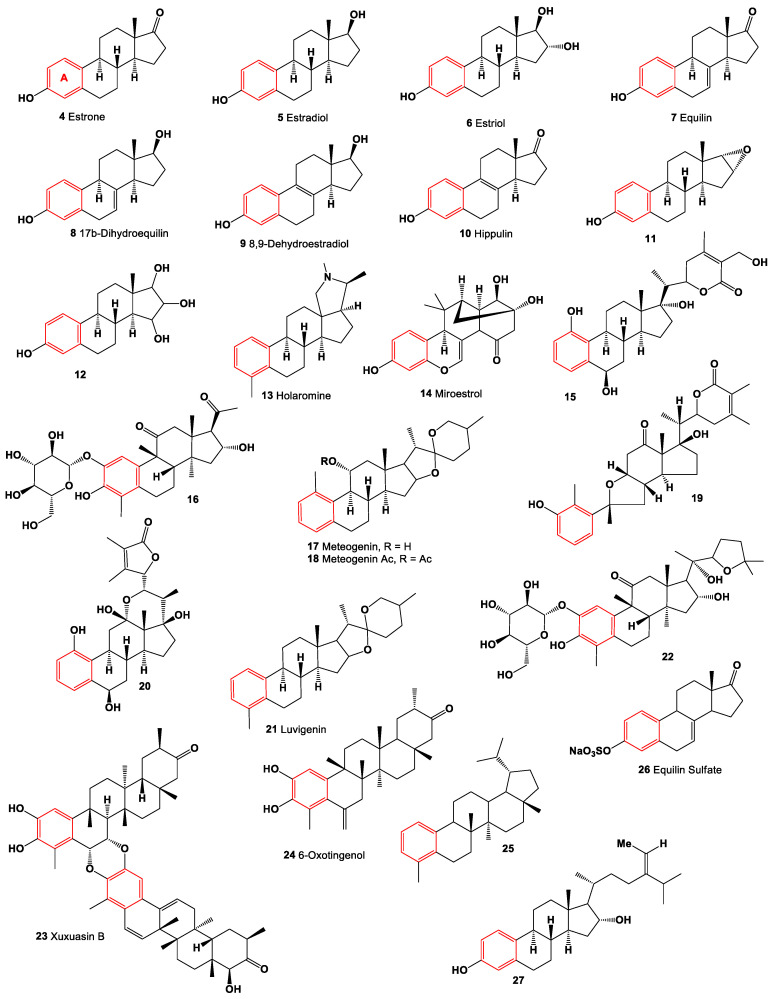
Steroids bearing ring A derived from plants.

**Figure 4 molecules-28-05549-f004:**
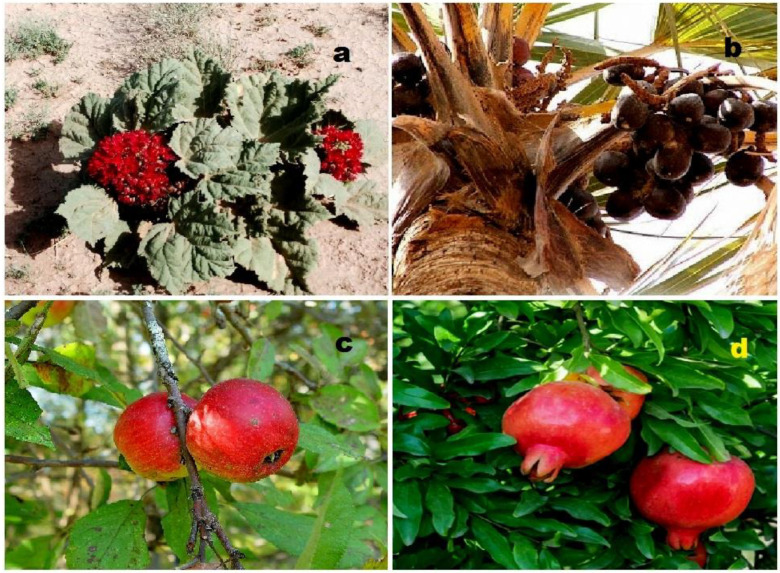
(**a**) *Glossostemon bruguieri*: *G. bruguieri* (also known as Moghat) is a shrub native to Iraq and Iran. In the past, it was cultivated in Egypt for its edible roots. The dried and peeled roots have been used in folk medicine to treat conditions such as gout and spasms and as a tonic and nourishment. Additionally, powdered Moghat has been traditionally consumed as a tonic and lactagogic remedy by women after childbirth. (**b**) *Hyphaene thebaica*: *H. thebaica* is a plant species commonly known as doum palm. It is native to regions of Africa and the Middle East. The seeds and pollen of *H. thebaica* are a source of estrone (**4**). (**c**) *Malus pumila*: *M. pumila*, commonly known as apple, is a fruit-bearing tree cultivated worldwide. Estrone (**4**) has been isolated from the seeds and pollen of *M. pumila*. (**d**) *Punica granatum*: *P. granatum*, or pomegranate, is a fruit-bearing shrub or small tree. It has been associated with various health benefits and may help prevent or treat conditions such as high blood pressure, high cholesterol, oxidative stress, hyperglycemia, and inflammatory activity. Estrone (**4**) has been found in *P. granatum*. Note: all photos used in this figure are obtained from sites where permission is granted for non-commercial use.

**Figure 5 molecules-28-05549-f005:**
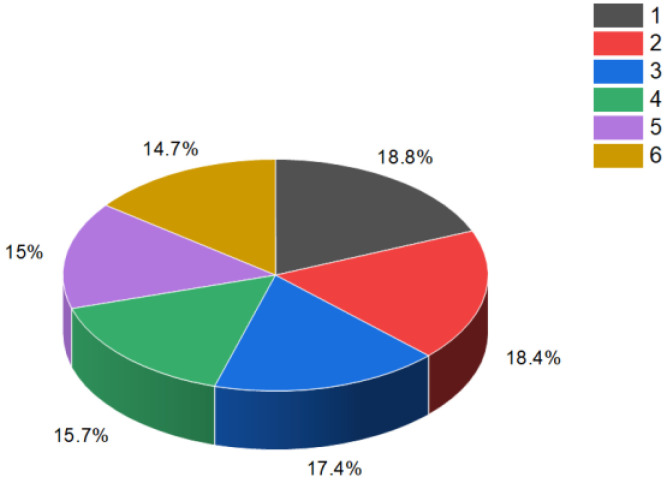
The percentage distribution of various biological activities associated with estrone (**4**), a compound known for its diverse pharmacological properties. The activities and their corresponding percentages are as follows: (1) ovulation inhibitor (18.8%); (2) cardiovascular analeptic (18.4%); (3) anti-hypercholesterolemic (17.4%); (4) apoptosis agonist (15.7%); (5) lipid metabolism regulator (15%); (6) prostate disorders treatment (14.7%). Estrone (**4**), which is a steroid bearing an aromatic ring A, is present in the pollen and seeds of numerous plants and plays a role in the reproductive development of these plants [35,36,37,38,39,40,41,42,43].

**Figure 6 molecules-28-05549-f006:**
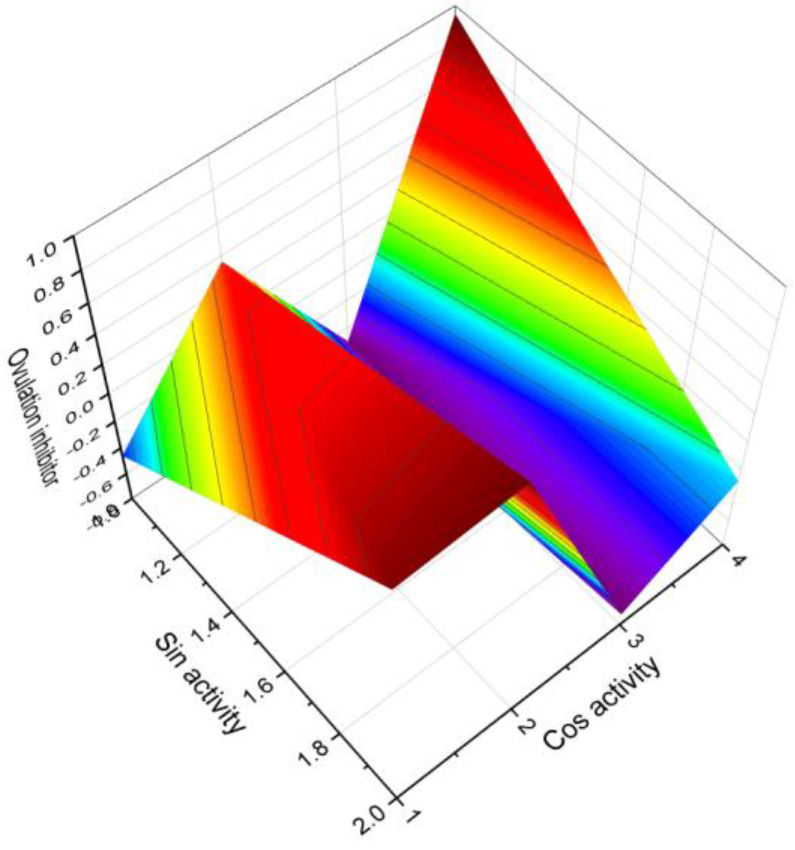
A 3D graph that illustrates the predicted and calculated activity of estrone (**4**, or estra-1,3,5(10)-triene-3-ol-17-one) as an ovulation inhibitor, with a confidence level exceeding 94%. This steroid has been isolated from the seeds and pollen of various plants, including *Glossostemon bruguieri*, *Hyphaene thebaica*, *Malus pumila*, *Phoenix dactylifera*, *Punica granatum*, and *Salix caprea*.

**Figure 7 molecules-28-05549-f007:**
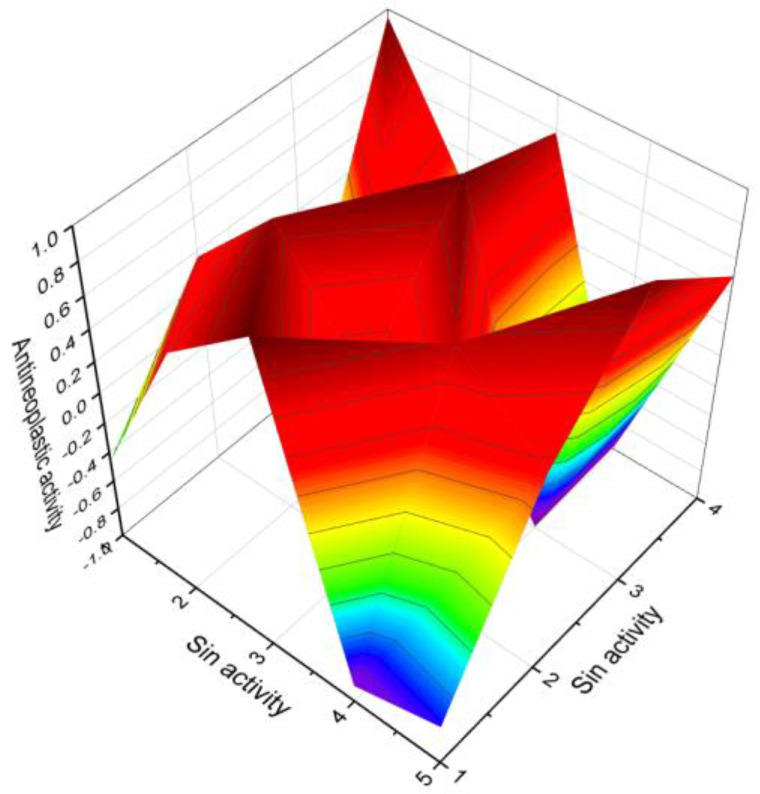
A 3D graph presenting the predicted and calculated anti-neoplastic activity of mono-aromatic ring A plant steroids (**16**, **17**, **21**, **23**, and **24**) with a confidence level exceeding 86%.

**Figure 8 molecules-28-05549-f008:**
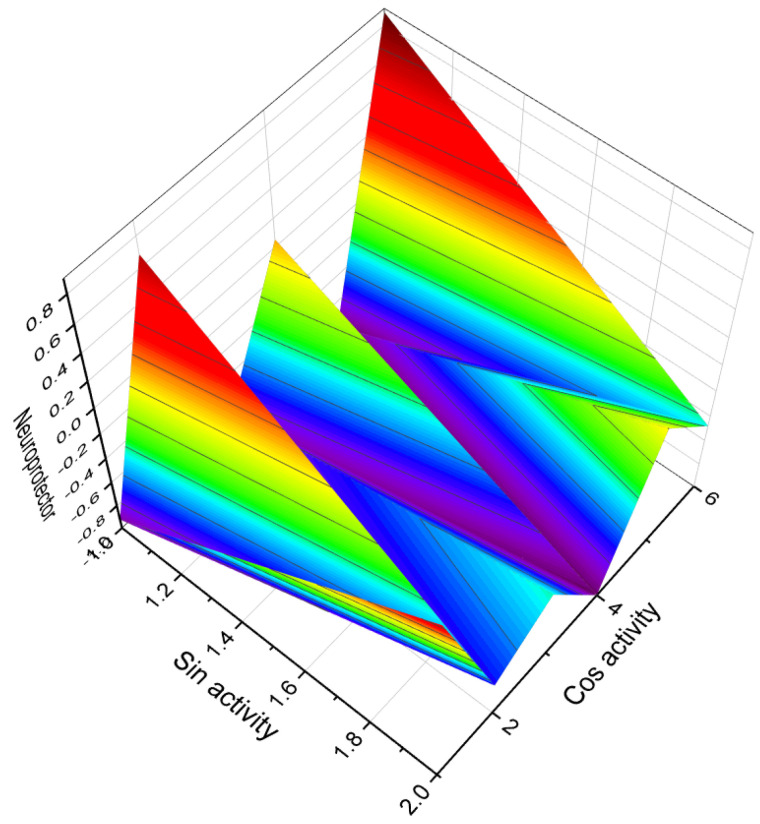
A 3D graph illustrating the predicted and calculated activity of the compound 4-hydroxy-6-oxopregnane-3-glycoside (**29**) as a neuroprotector. The graph demonstrates the relationship between the compound’s structure and its predicted neuroprotective activity with a confidence level exceeding 97%. This steroid, containing an aromatic ring A, was isolated from a Pohnpei sponge known as *Cribrochalina olemda*. The graph provides insight into the relationship between the molecular structure of the compound and its predicted efficacy as a neuroprotector. By analyzing the graph, one can observe how variations in the structural features of the compound may impact its potential neuroprotective effects. The high confidence level of over 97% suggests a strong reliability in the predicted activity of this steroid as a neuroprotector. Understanding the neuroprotective activity of compounds is crucial for the development of potential treatments or interventions for neurodegenerative disorders, brain injuries, and other conditions that affect the health and function of the nervous system. Neuroprotector activity refers to the ability of a compound to protect and preserve the health and function of neurons in the brain and nervous system.

**Figure 9 molecules-28-05549-f009:**
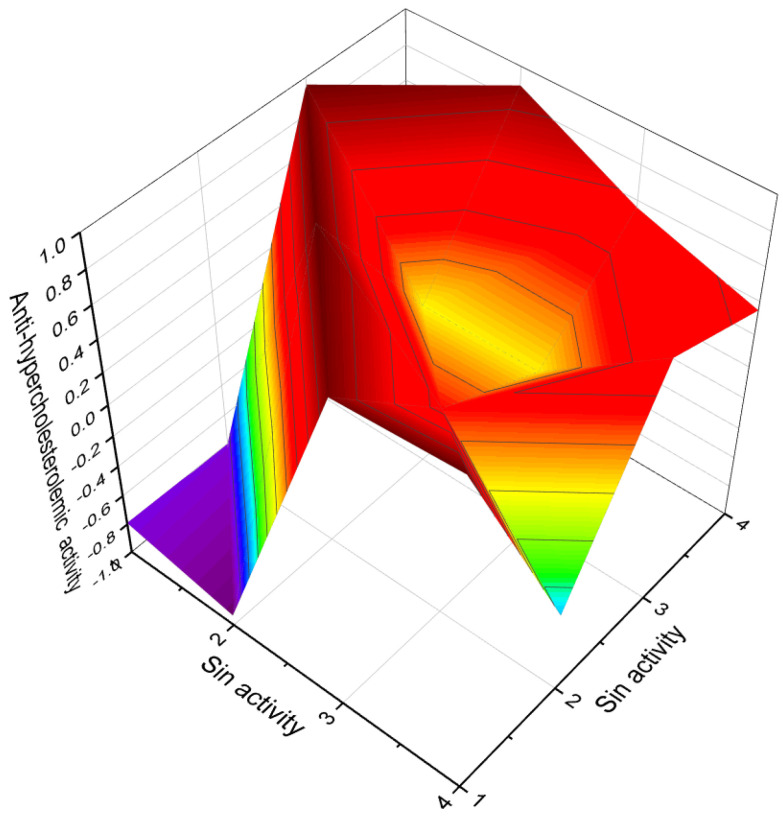
A 3D graph that illustrates the predicted and calculated anti-hypercholesterolemic activity of mono-aromatic ring A plant steroids (33, 34, 35, and 37) with a confidence level exceeding 91%. Anti-hypercholesterolemic activity refers to the ability of a compound to help reduce high cholesterol levels in the body. The graph showcases the relationship between the chemical structures of these mono-aromatic ring A plant steroids and their predicted efficacy in combating hypercholesterolemia. By analyzing the graph, it is possible to observe the correlation between the structural features of these compounds and their potential anti-hypercholesterolemic effects. The confidence level of over 91% indicates a high degree of reliability in the predicted activity of these steroids.

**Figure 10 molecules-28-05549-f010:**
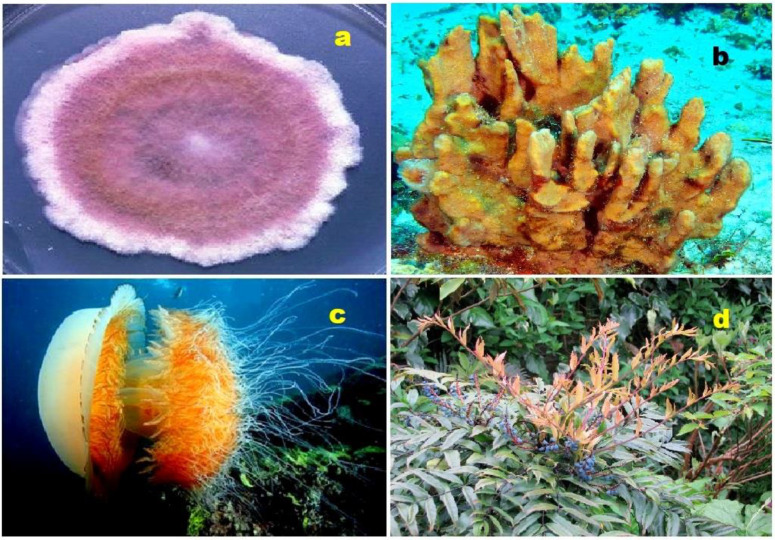
Steroid (**35**) was found in the mycelium of pathogenic fungus *Fusarium roseum* (**a**); steroid (**37**) was isolated from a marine sponge *Topsentia* sp. (**b**); steroid (**38**) is a metabolite from the giant jellyfish *Nemopilema nomurai* (**c**); and steroid (**39**) is produced by an endophytic fungus, *Diaporthe* sp., which inhabits leaves of the Chinese medicinal plant *Mahonia fortunei* (**d**).

**Figure 11 molecules-28-05549-f011:**
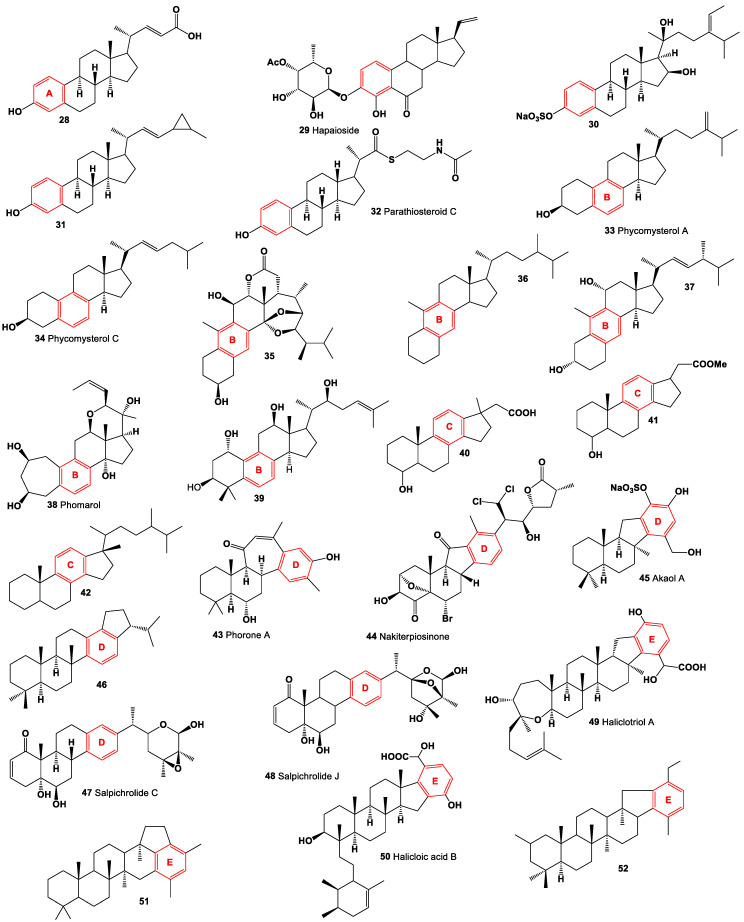
Steroids bearing ring A, B, C, D, and E in natural sources.

**Figure 12 molecules-28-05549-f012:**
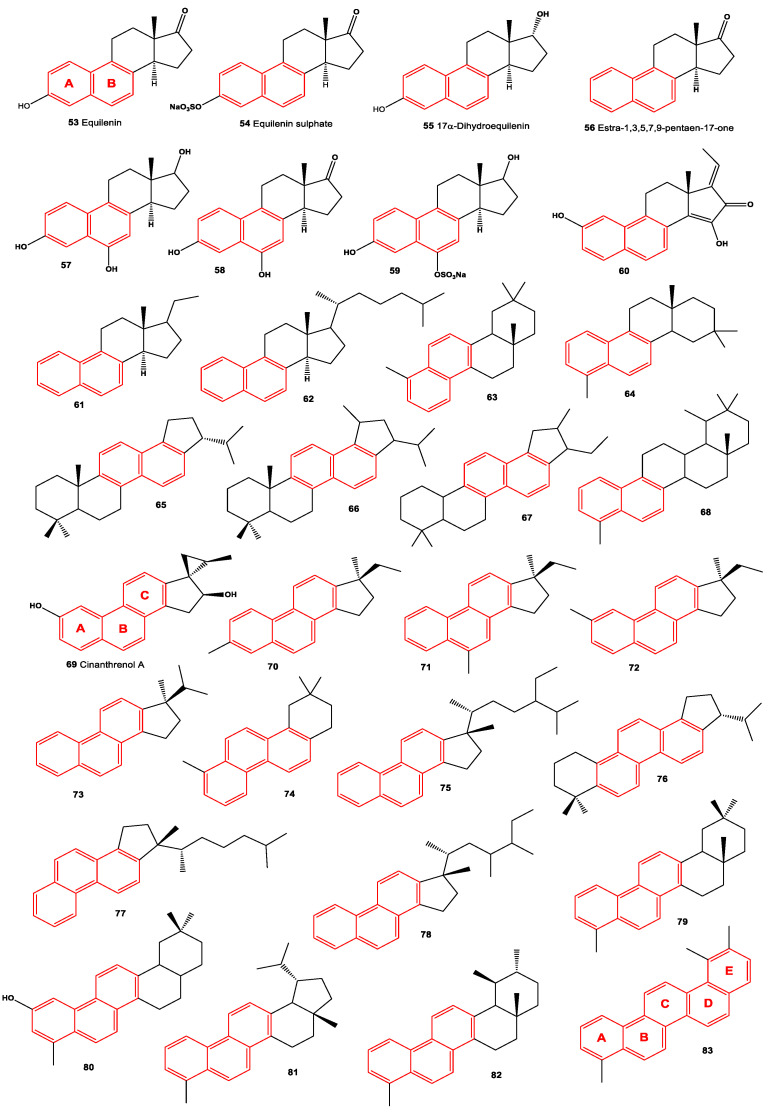
Di- and tri-aromatic steroids and terpenoids derived from fungi, invertebrates, sediments, and petroleum.

**Figure 13 molecules-28-05549-f013:**
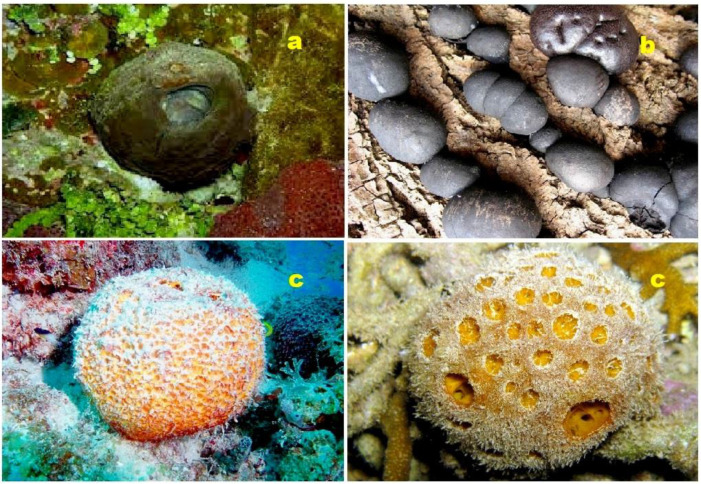
Di-aromatic steroid (**60**) was found in the marine sponge *Strongylophora* sp. (**a**); another di-aromatic steroid (**62**) was produced by the ascomycete *Daldinia concentrica* (**b**); and tri-aromatic steroids or phenanthrene-containing steroids (**69**–**73**) were found in the marine sponge *Cinachyrella* sp. (**c**).

**Figure 14 molecules-28-05549-f014:**
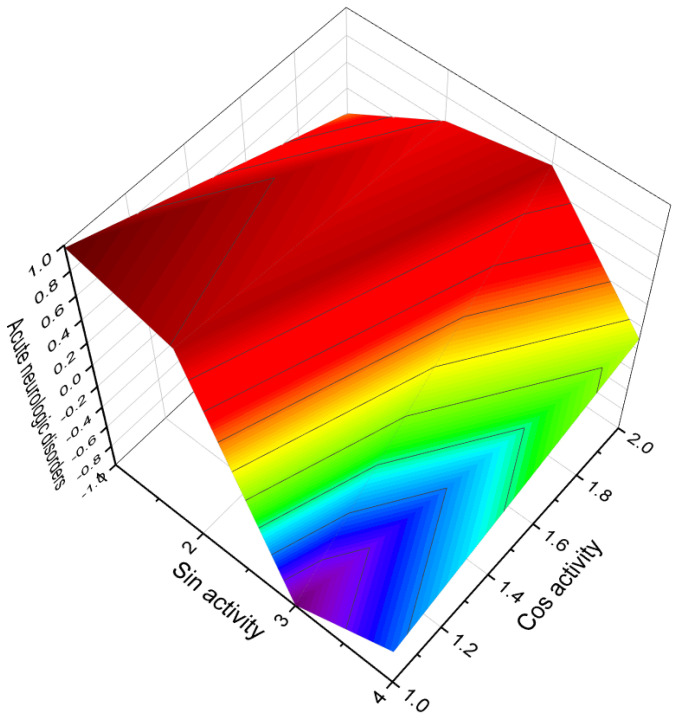
A 3D graph showing the predicted and calculated activity as a treatment for acute neurological disorders of aromatic steroid (**81**) with over 92% confidence.

**Figure 15 molecules-28-05549-f015:**
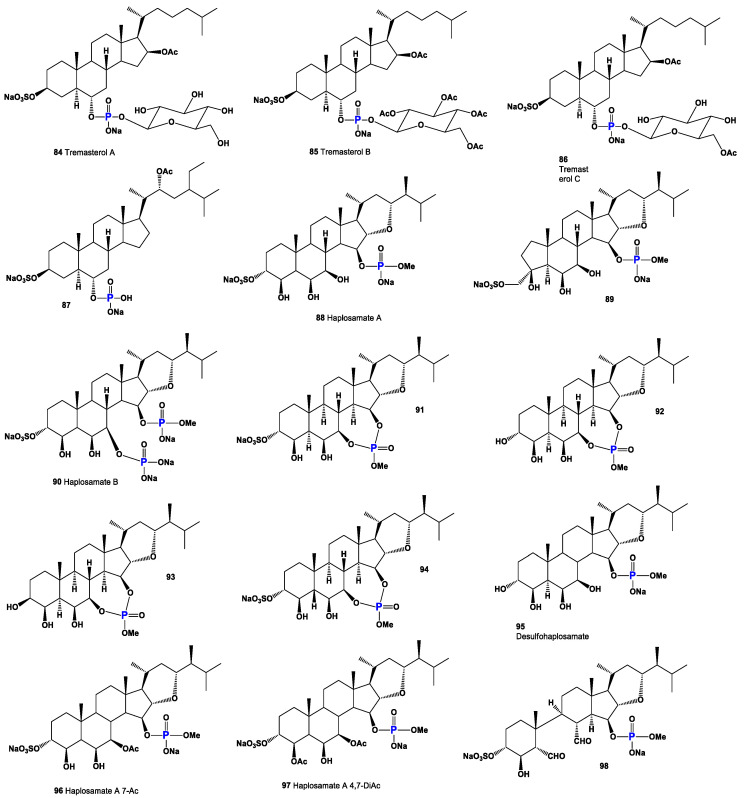
Natural bioactive steroid phosphate esters.

**Figure 16 molecules-28-05549-f016:**
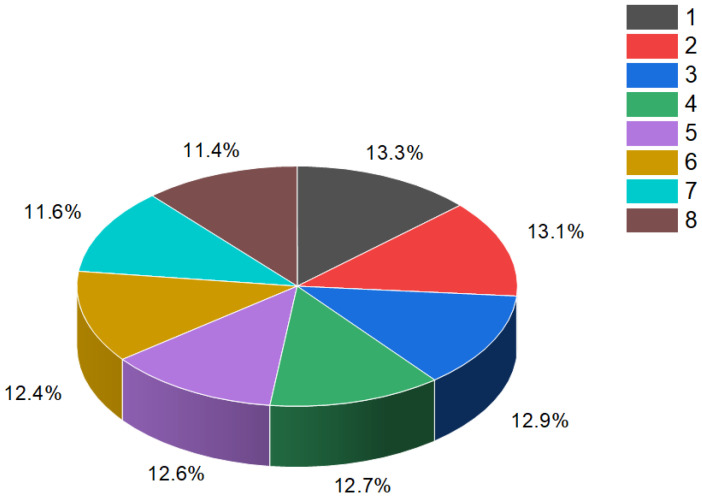
Illustration of the percentage distribution of biological activities on the example of tremasterol A (**84**), which is derived from the marine starfish *Tremaster novaecaledoniae*, and this steroid has a wide range of pharmacological properties. Activities are indicated under the numbers: 1, wound-healing agent (13.3%); 2, hepatoprotectant (13.1%); 3, analeptic (12.9%); 4, laxative (12.7%); 5, anti-hypercholesterolemic (12.6%); 6, anti-carcinogenic (12.4%); 7, hemostatic (11.6%), and 8, anti-neoplastic (11.4%).

**Figure 17 molecules-28-05549-f017:**
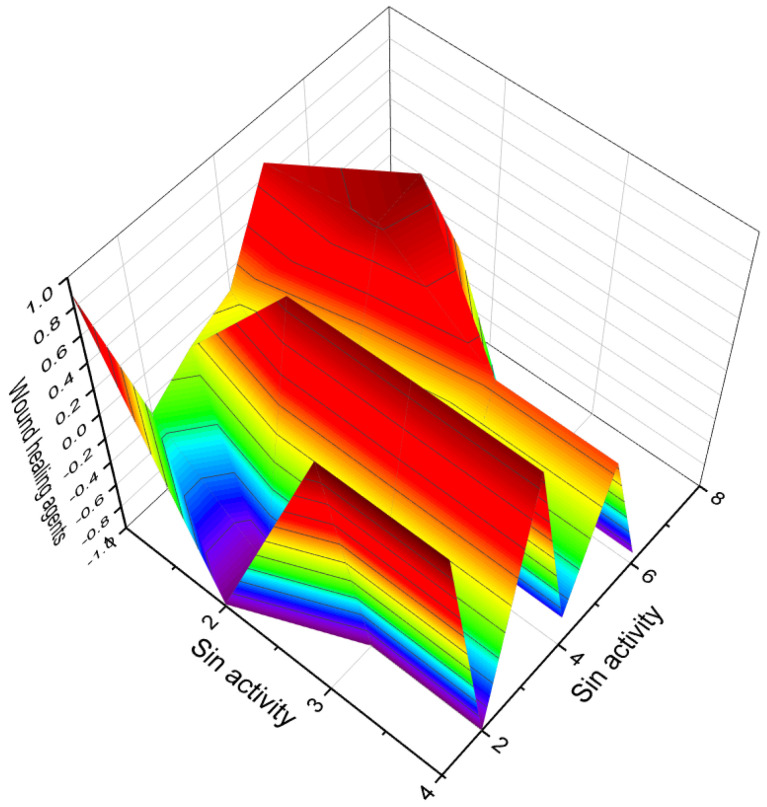
A 3D graph showing the predicted and calculated activity as wound-healing agents of steroid phosphate esters (**84**, **86**, **87**, and **88**) with over 89% confidence. Wound-healing agents are substances or treatments that promote the healing of wounds. These agents can be in the form of medications, dressings, or therapies that aid in the different stages of the wound-healing process. The wound-healing process involves a series of complex biological events that aim to restore the damaged tissue and close the wound. The stages of wound healing include hemostasis (stopping bleeding), inflammation, proliferation, and remodeling.

**Figure 18 molecules-28-05549-f018:**
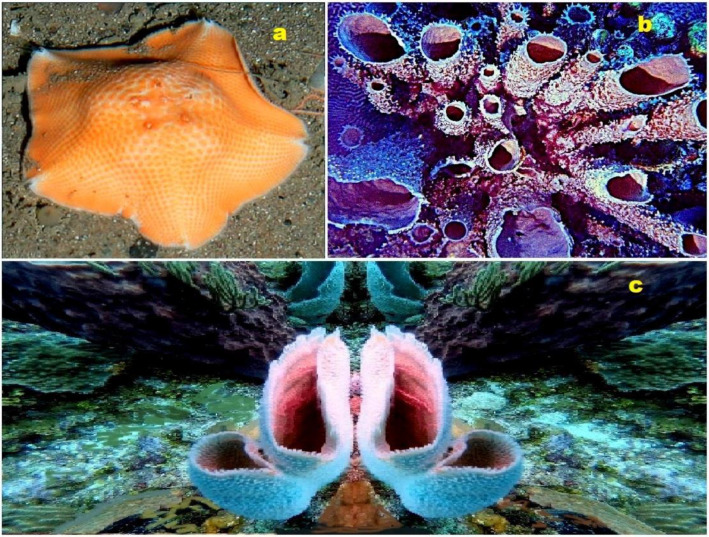
Steroid phosphate esters (**84**–**87**) were isolated from the deep marine starfish *Tremaster novaecaledoniae* (**a**); steroids (**88**), (**89**), and (**90**) are found in a marine sponge *Cribrochalina* sp. (**b**); and steroids (**88** and **90**) were also found in the Indonesian sponge *Dasychalina* sp. (**c**).

**Figure 19 molecules-28-05549-f019:**
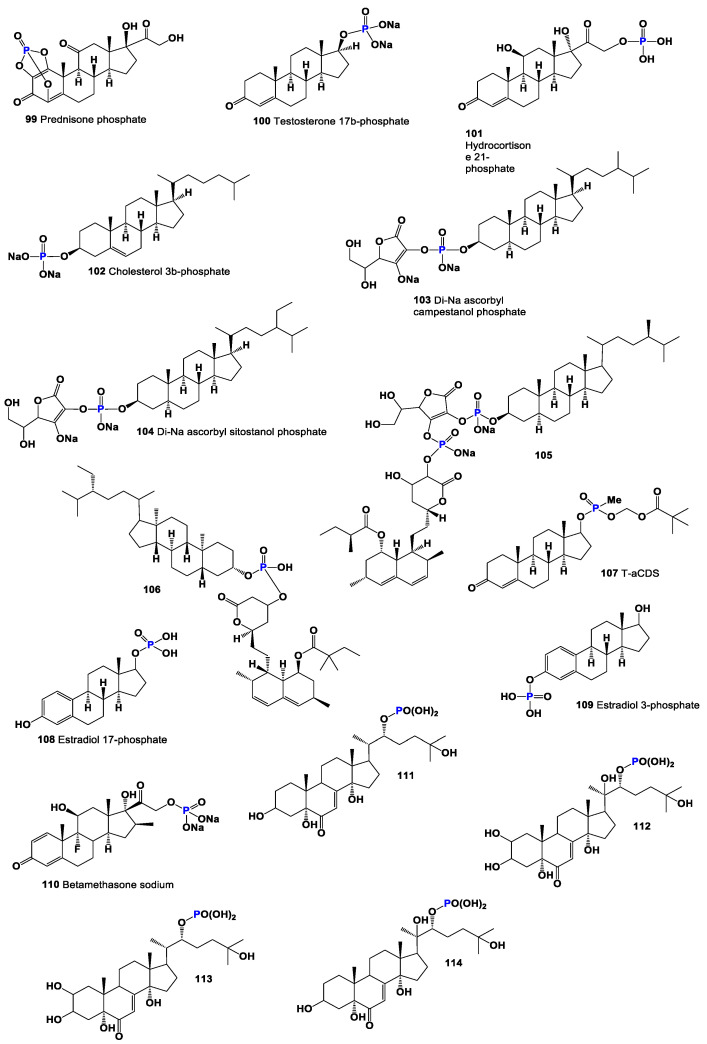
Natural and semi-synthetic bioactive steroid phosphate esters.

**Figure 23 molecules-28-05549-f023:**
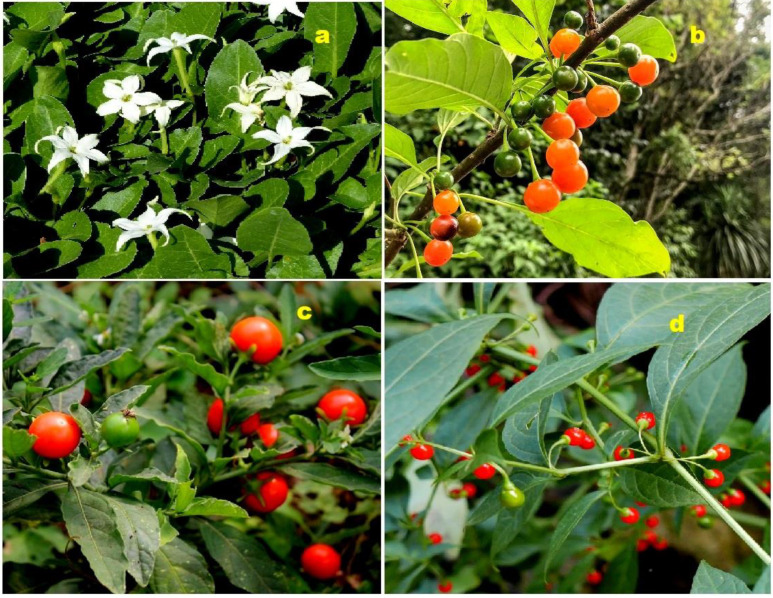
Various plant species wherein sterols containing a chlorine atom have been discovered. Chlorinated steroids (**115** and **116**) were isolated from the leaves of the *Jaborosa integrifolia* (**a**); steroids (**116**) and (**117**) were found in *Acnistus breviflorus* (**b**); withanolide Z (**129**) was isolated from *Withania somnifera* (**c**); and steroid (**140**) was found in *Tubocapsicum anomalum* (**d**).

**Figure 24 molecules-28-05549-f024:**
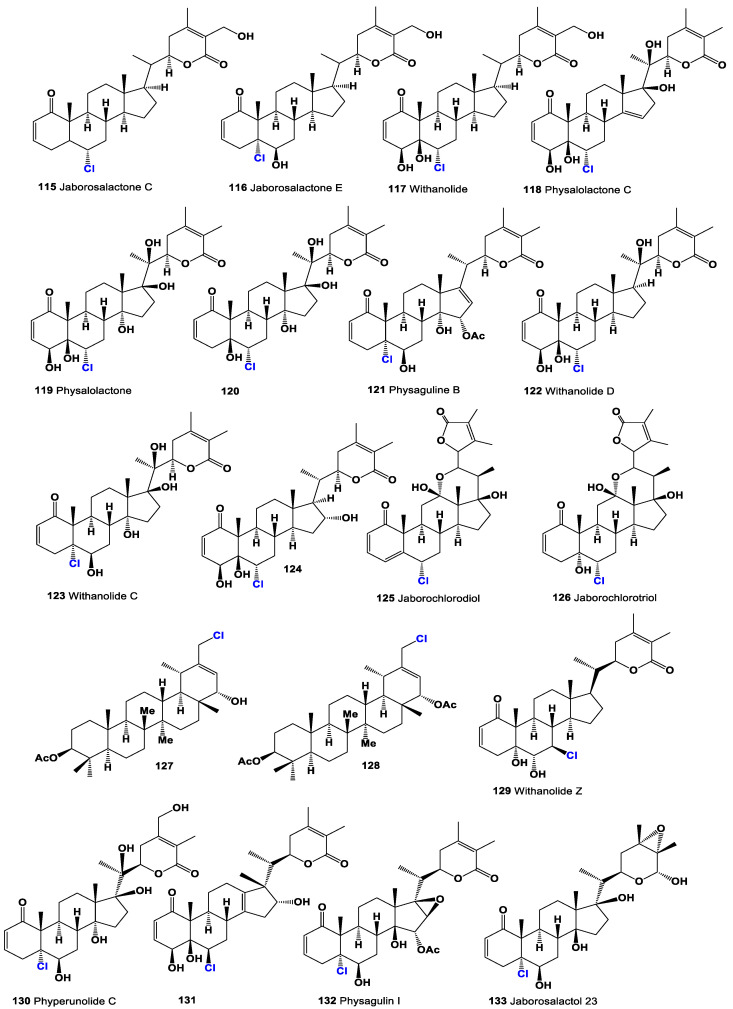
Chlorinated steroids and triterpenoids derived from plants.

**Figure 25 molecules-28-05549-f025:**
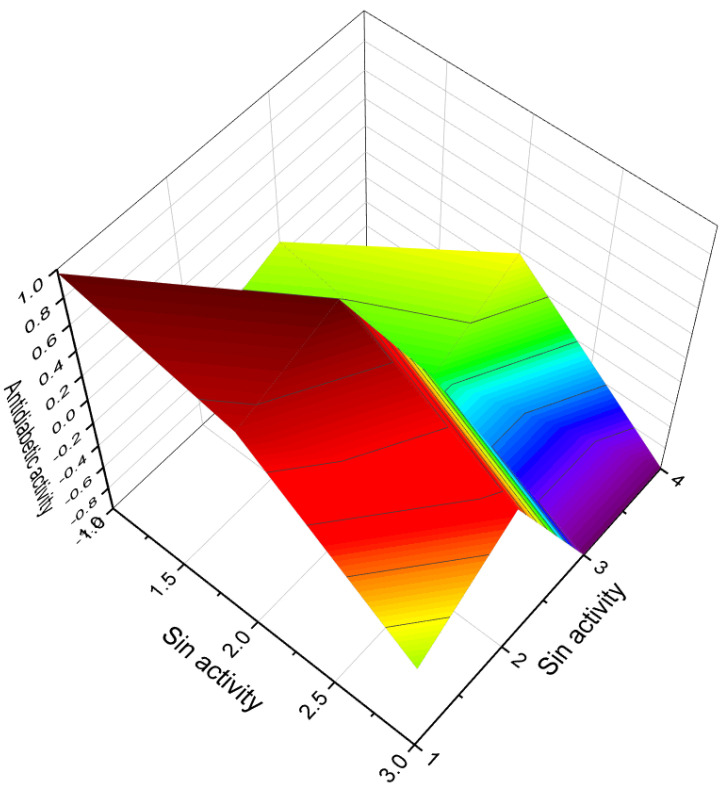
A 3D graph displaying the predicted and calculated anti-diabetic activity of chlorinated steroids, specifically compounds **118**, **119**, and **120**. The graph provides insights into the relationship between the activity of these compounds and their potential efficacy in managing diabetes. Anti-diabetic activity refers to the ability of a compound to help manage or control diabetes, a metabolic disorder characterized by high blood sugar levels. Compounds with anti-diabetic activity can exert various effects on glucose metabolism, insulin sensitivity, and other related pathways. The predicted and calculated activity values depicted on the graph represent the potency or effectiveness of the chlorinated steroids in terms of their anti-diabetic properties. With a confidence level of over 94%, the graph indicates a high degree of certainty in the accuracy of the predicted and calculated activity values. The exploration of chlorinated steroids for their anti-diabetic activity is of great interest in the field of diabetes research. Identifying compounds with potential anti-diabetic properties can contribute to the development of new treatment approaches and therapies for individuals living with diabetes. It is important to note that further research, including in vitro and in vivo studies, is necessary to fully understand the mechanisms of action, optimal dosage, and potential applications of these chlorinated steroids in managing diabetes. Additionally, clinical trials would be required to assess their safety and efficacy in human subjects. The study of chlorinated steroids and their anti-diabetic activity holds promise in advancing our understanding of natural compounds that may help in the management of diabetes and related metabolic disorders.

**Figure 26 molecules-28-05549-f026:**
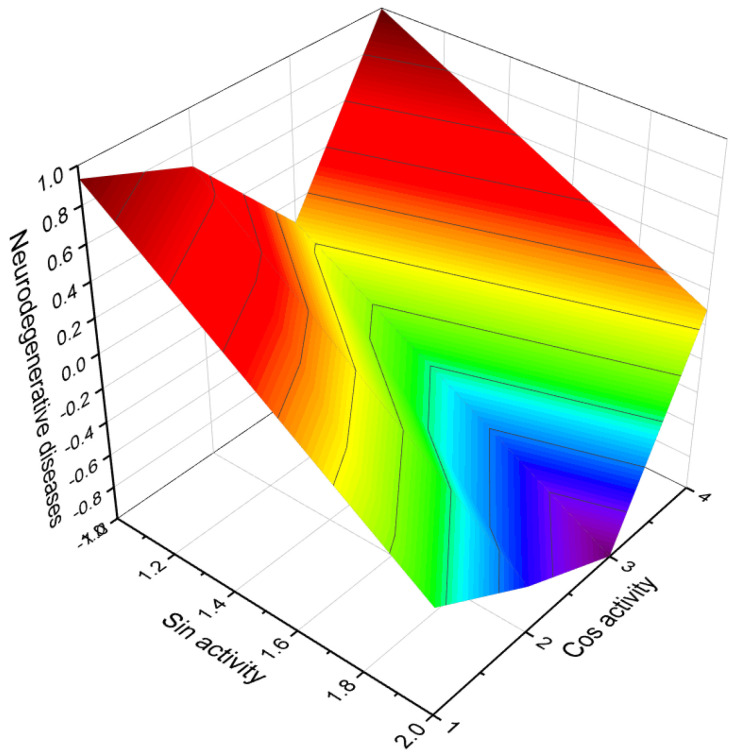
A 3D graph depicting the predicted and calculated activity of a specific chlorinated steroid (**122**) as a potential treatment for neurodegenerative diseases, specifically Alzheimer’s and Parkinson’s diseases. The graph provides insights into the relationship between the activity of the compound and its potential efficacy in treating these conditions. Neurodegenerative diseases, such as Alzheimer’s disease and Parkinson’s disease, are characterized by the progressive loss of structure and function of neurons in the central nervous system. These diseases often lead to cognitive decline, motor impairments, and various neurological symptoms. The predicted and calculated activity values shown on the graph represent the potency or effectiveness of the chlorinated steroid in terms of its activity against neurodegenerative diseases. With a confidence level of over 92%, the graph indicates a high degree of certainty in the accuracy of the predicted and calculated activity values. The exploration of chlorinated steroids for their potential therapeutic effects in neurodegenerative diseases is an important area of research. These compounds may interact with various molecular targets and pathways involved in the pathogenesis of these diseases, potentially slowing down or preventing neuronal degeneration, reducing inflammation, or promoting neuroprotective mechanisms. It is crucial to note that further research, including preclinical and clinical studies, is necessary to fully understand the mechanisms of action, therapeutic potential, and safety profile of the specific chlorinated steroid (**122**) and other compounds in the treatment of neurodegenerative diseases. Developing effective treatments for Alzheimer’s and Parkinson’s diseases remains a significant challenge, and ongoing research is vital in advancing our understanding and finding novel therapeutic strategies. The study of chlorinated steroids and their potential role in neurodegenerative diseases provides hope for the development of new therapeutic interventions that can improve the quality of life for individuals affected by these devastating conditions.

**Figure 27 molecules-28-05549-f027:**
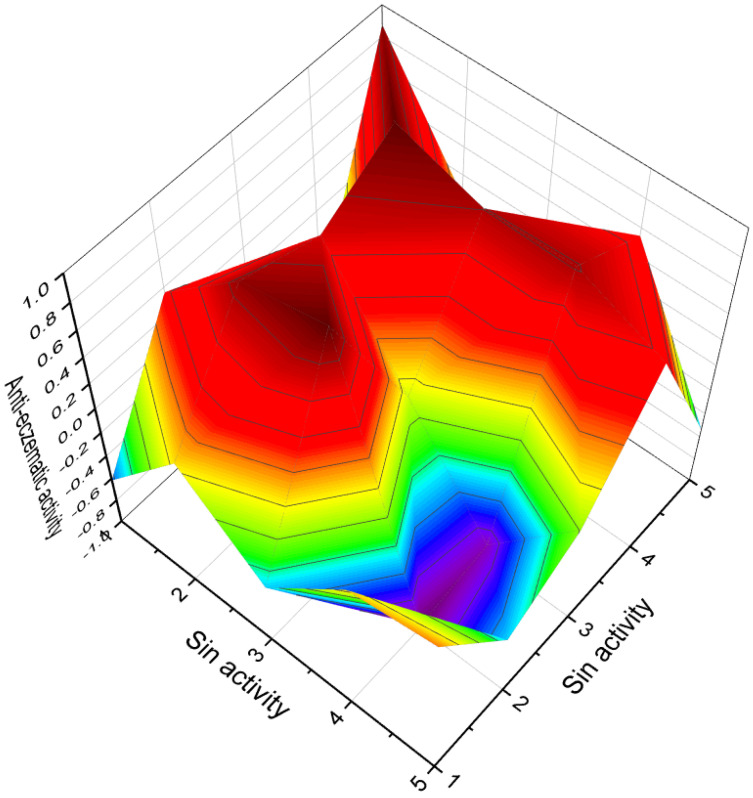
A 3D graph illustrating the predicted and calculated anti-eczematic activity of chlorinated steroids, specifically compounds **115**, **116**, **117**, **124**, and **132**. The graph provides insights into the relationship between the activity of these compounds and their potential efficacy in treating eczema. Anti-eczematic activity refers to the ability of a compound to alleviate or manage symptoms associated with eczema, a chronic inflammatory skin condition characterized by itching, redness, and rash. Compounds with anti-eczematic activity can help reduce inflammation, relieve itching, and promote skin healing. The predicted and calculated activity values depicted on the graph represent the potency or effectiveness of the chlorinated steroids in terms of their anti-eczematic properties. With a confidence level of over 91%, the graph indicates a high degree of certainty in the accuracy of the predicted and calculated activity values. The exploration of chlorinated steroids for their anti-eczematic activity holds promise in the field of dermatology and skin health. Identifying compounds that can effectively reduce inflammation, alleviate itching, and promote skin repair can significantly improve the management of eczema. It is important to note that further research, including in vitro and clinical studies, is necessary to fully understand the mechanisms of action, optimal dosage, and potential applications of these chlorinated steroids in treating eczema. Additionally, comprehensive safety evaluations would be required to assess their suitability for use in human subjects.

**Figure 28 molecules-28-05549-f028:**
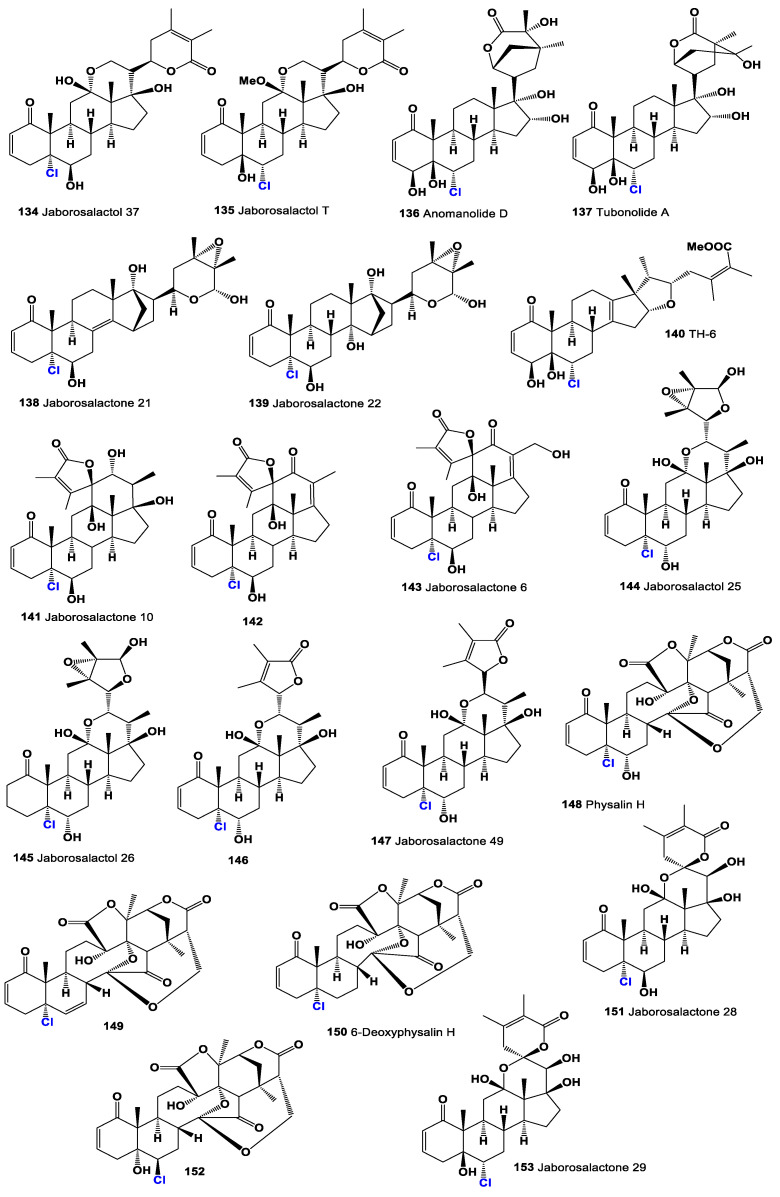
Chlorinated highly oxygenated steroids derived from plants.

**Figure 29 molecules-28-05549-f029:**
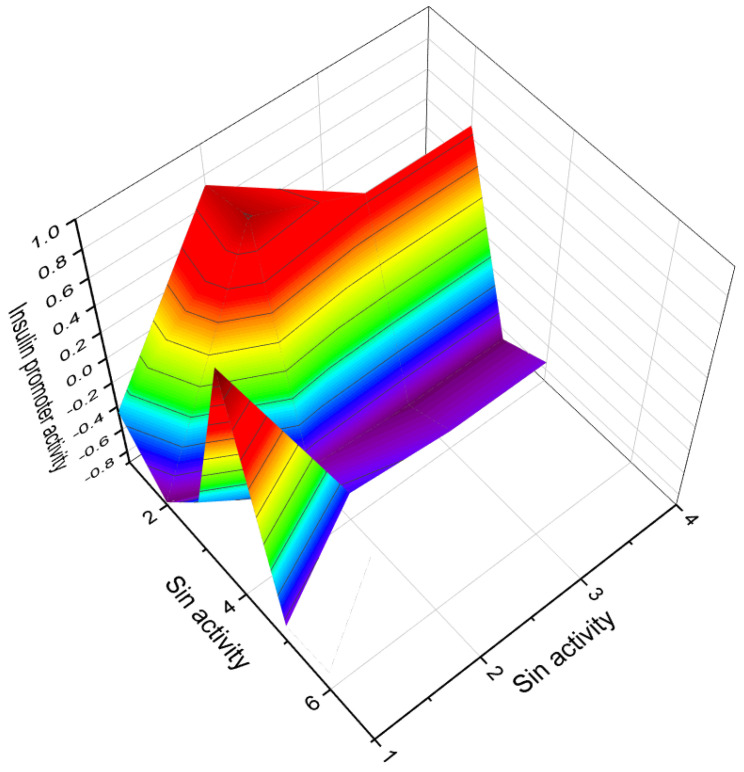
A 3D graph illustrating the predicted and calculated insulin promoter activity of chlorinated steroids, specifically compounds **131**, **135**, **136**, and **137**. The graph provides insights into the relationship between the activity of these compounds and their potential efficacy in promoting insulin production. Insulin promoter activity refers to the ability of a compound to enhance the production or secretion of insulin, a hormone that plays a crucial role in regulating blood sugar levels. Compounds with insulin promoter activity can help improve glucose metabolism and enhance insulin signaling, which is beneficial for individuals with conditions such as diabetes. The predicted and calculated activity values displayed on the graph represent the potency or effectiveness of the chlorinated steroids in terms of their insulin promoter properties. With a confidence level of over 98%, the graph indicates a high degree of certainty in the accuracy of the predicted and calculated activity values. The exploration of chlorinated steroids for their insulin promoter activity holds significant promise in the field of diabetes research. Identifying compounds that can enhance insulin production or secretion can contribute to the development of new strategies for managing diabetes and improving glycemic control.

**Figure 30 molecules-28-05549-f030:**
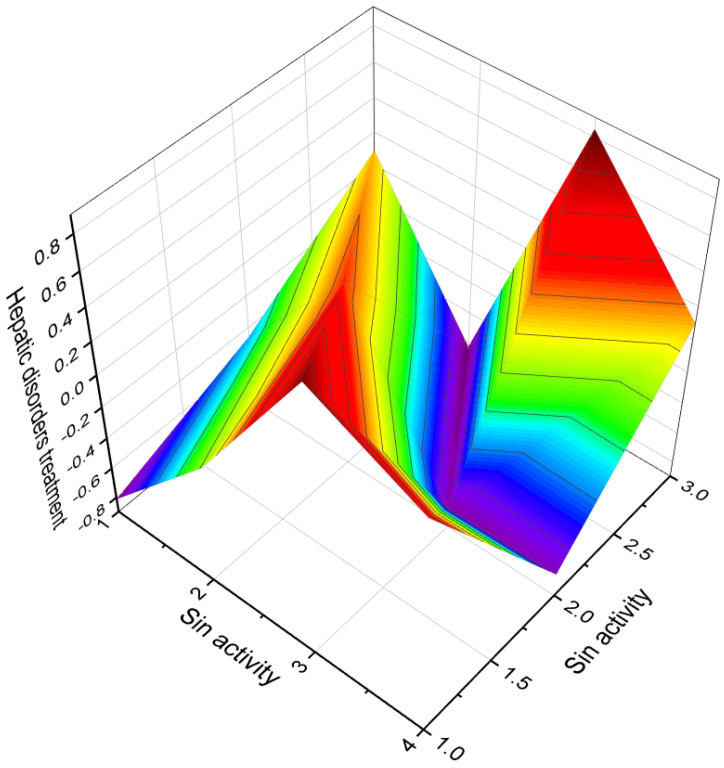
A 3D graph depicting the predicted and calculated activity of chlorinated steroids, specifically compounds **141**, **142**, and **143**, as potential treatments for liver disease. The graph provides insights into the relationship between the activity of these compounds and their potential efficacy in treating liver diseases. Liver disease refers to a wide range of conditions that affect the liver, impairing its normal functioning. These conditions can include liver inflammation (hepatitis), fatty liver disease, cirrhosis, liver cancer, and others. Treatment options for liver disease are diverse, including medications that can help manage symptoms, slow down disease progression, or promote liver regeneration. The predicted and calculated activity values displayed on the graph represent the potency or effectiveness of the chlorinated steroids in terms of their activity against liver disease. With a confidence level of over 93%, the graph indicates a high degree of certainty in the accuracy of the predicted and calculated activity values. The exploration of chlorinated steroids for their potential therapeutic effects in liver disease is an area of active research. These compounds may interact with various molecular targets and pathways involved in liver function, inflammation, and regeneration, potentially offering benefits in the management of liver diseases.

**Figure 31 molecules-28-05549-f031:**
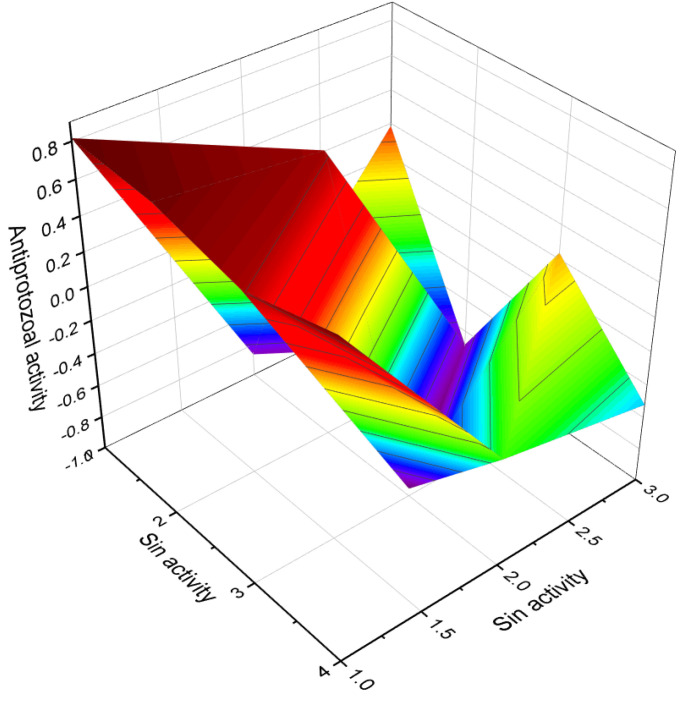
A 3D graph illustrating the predicted and calculated anti-protozoal activity of chlorinated steroids, specifically compounds **148**, **149**, and **150**. The graph provides insights into the relationship between the activity of these compounds and their potential efficacy in inhibiting protozoan parasites. Anti-protozoal activity refers to the ability of a compound to inhibit the growth or survival of protozoan parasites, which are single-celled organisms that can cause various infectious diseases in humans and animals. Protozoan parasites can cause diseases such as malaria, leishmaniasis, trypanosomiasis, and toxoplasmosis, among others. The predicted and calculated activity values depicted on the graph represent the potency or effectiveness of the chlorinated steroids in terms of their anti-protozoal properties. With a confidence level of over 95%, the graph indicates a high degree of certainty in the accuracy of the predicted and calculated activity values. The exploration of chlorinated steroids for their anti-protozoal activity is of great interest in the field of parasitology and drug discovery. Identifying compounds that can effectively target and inhibit protozoan parasites can lead to the development of new treatments for various protozoal infections.

**Figure 32 molecules-28-05549-f032:**
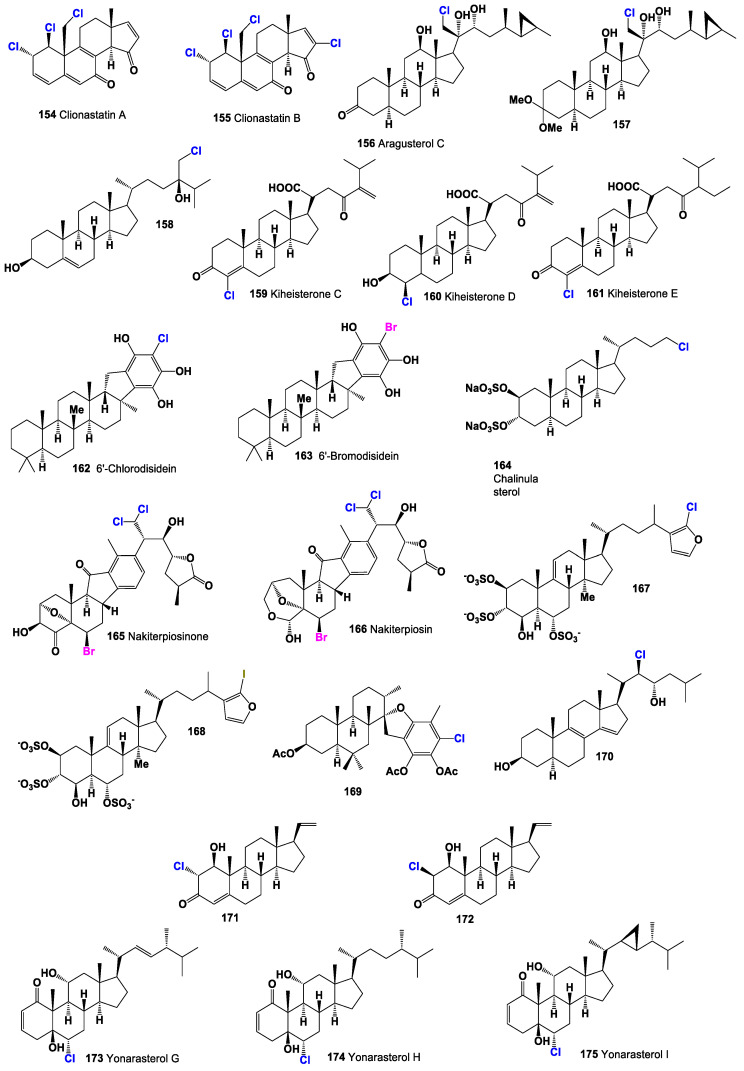
Halogenated steroids derived from marine sources.

**Figure 33 molecules-28-05549-f033:**
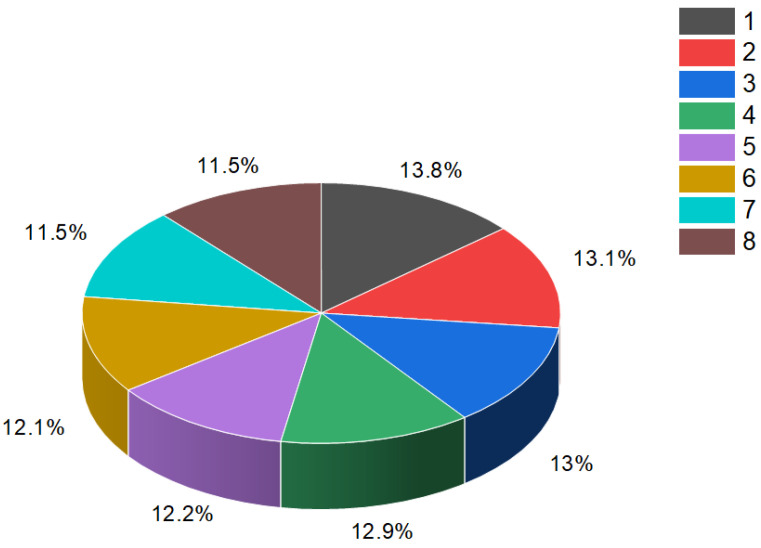
The percentage distribution of biological activities on the example of chlorinated steroid aragusterol C (**156**) from marine sponge *Xestospongia* sp., which has a wide range of pharmacological properties. The found activities are indicated under the numbers: 1, anti-neoplastic (13.8%); 2, prostate cancer treatment (13.1%); 3, anti-sarcoma cancer (13%); 4, anti-renal cancer (12.9%); 5, choleretic (12.2%); 6, anti-hypercholesterolemic (12.1%); 7, anti-fungal (11.5%); and 8, dermatologic (11.5%).

**Figure 34 molecules-28-05549-f034:**
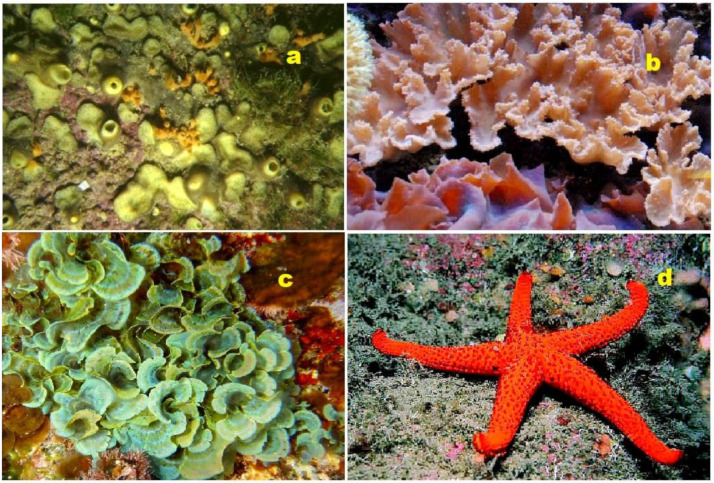
Some of the marine invertebrates and algae that produce chlorinated steroids. Chlorinated steroids (**154**) and B (**155**) were found in the sponge *Cliona nigricans* (**a**); chlorinated steroid (**158**) was isolated from the soft coral *Sinularia brassica* (**b**); the brown alga *Stypopodium flabelliforme* contained steroid (**169**) (**c**), and steroid (**170**) was found in starfish *Echilaster sepositus* (**d**).

**Figure 35 molecules-28-05549-f035:**
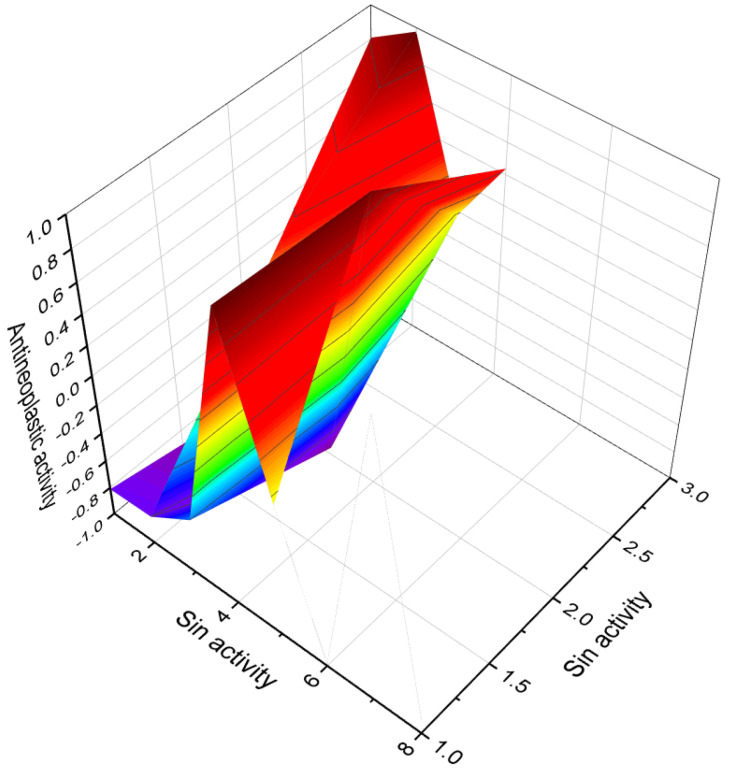
A 3D graph illustrating the predicted and calculated anti-neoplastic activity of halogenated steroids (**156**, **157**, and **171**) derived from marine sources. Anti-neoplastic activity refers to the ability of a substance to inhibit or prevent the growth and spread of cancer cells. It is a crucial characteristic for potential cancer treatments. The graph provides valuable insights into the relationship between the structural features of these halogenated steroids and their potential effectiveness as anti-neoplastic agents. The predicted and calculated activity values depicted on the graph represent the potency or efficacy of these compounds for inhibiting the growth and proliferation of cancer cells. With a confidence level of over 92%, the graph indicates a high degree of certainty in the accuracy of the predicted and calculated anti-neoplastic activity values. This suggests that these halogenated steroids hold promise as potential candidates for further investigation as anti-cancer agents. The exploration of halogenated steroids derived from marine sources for their anti-neoplastic activity is of great interest in cancer research and drug development. Natural compounds with anti-neoplastic properties offer potential alternatives or adjuncts to traditional cancer therapies.

**Table 1 molecules-28-05549-t001:** Biological activities of mono-aromatic steroids (**4**–**27**).

No.	Dominated Biological Activity (Pa) *	Additional Predicted Activities (Pa) *
**4**	Ovulation inhibitor (0.942) Cardiovascular analeptic (0.924) Apoptosis agonist (0.750)	Anti-hypercholesterolemic (0.871) Lipid metabolism regulator (0.788) Prostate disorders treatment (0.737)
**5**	Anti-hypercholesterolemic (0.894) Ovulation inhibitor (0.889) Anesthetic general (0.868)	Respiratory analeptic (0.851) Acute neurologic disorders treatment (0.793) Prostate disorders treatment (0.729)
**6**	Anesthetic general (0.845) Ovulation inhibitor (0.832)	Acute neurologic disorders treatment (0.822) Neuroprotector (0.815)
**7**	Anti-hypercholesterolemic (0.856) Ovulation inhibitor (0.847) Cardiovascular analeptic (0.842)	Lipid metabolism regulator (0.788) Apoptosis agonist (0.750) Prostate disorders treatment (0.725)
**8**	Anti-hypercholesterolemic (0.885) Apoptosis agonist (0.801)	Hepatic disorders treatment (0.739) Ovulation inhibitor (0.726)
**9**	Acute neurologic disorders treatment (0.871) Respiratory analeptic (0.843) Vasoprotector (0.811)	Neuroprotector (0.785) Anesthetic general (0.753) Ovulation inhibitor (0.740)
**10**	Cardiovascular analeptic (0.882) Ovulation inhibitor (0.860)	Respiratory analeptic (0.846) Acute neurologic disorders treatment (0.844)
**11**	Respiratory analeptic (0.879) Ovulation inhibitor (0.765)	Neuroprotector (0.762) Cardiovascular analeptic (0.692)
**12**	Acute neurologic disorders treatment (0.849) Vasoprotector (0.795)	Anti-inflammatory (0.788) Ovulation inhibitor (0.778)
**13**	Psychotropic (0.815) Ovulation inhibitor (0.586)	Attention deficit/hyperactivity disorder treatment (0.744)
**14**	Postmenopausal disorders treatment (0.945)	Anti-inflammatory (0.669)
**15**	Lipid metabolism regulator (0.913) Cytostatic (0.891) Anti-neoplastic (0.876)	Hepatoprotectant (0.845) Immunosuppressant (0.792) Apoptosis agonist (0.784)
**16**	Chemopreventive (0.919) Proliferative diseases treatment (0.914)	Anti-neoplastic (0.837) Vasoprotector (0.824)
**17**	Apoptosis agonist (0.893) Anti-neoplastic (0.827)	Anti-inflammatory (0.873) Hypolipemic (0.854)
**18**	Apoptosis agonist (0.883) Anti-neoplastic (0.826)	Hypolipemic (0.863) Anti-inflammatory (0.855)
**19**	Anti-neoplastic (0.879) Apoptosis agonist (0.775)	Immunosuppressant (0.744) Anti-inflammatory (0.715)
**20**	Anti-neoplastic (0.782)	Genital warts treatment (0.736)
**21**	Apoptosis agonist (0.896) Anti-neoplastic (0.843)	Hypolipemic (0.850) Anti-inflammatory (0.814)
**22**	Chemopreventive (0.887) Anti-neoplastic (0.794)	Anti-inflammatory (0.819) Proliferative diseases treatment (0.784)
**23**	Anti-neoplastic (0.909) Apoptosis agonist (0.790)	Anti-inflammatory (0.822) Immunosuppressant (0.727)
**24**	Anti-neoplastic (0.888) Apoptosis agonist (0.847)	Anti-inflammatory (0.830) Immunosuppressant (0.739)
**25**	Anti-neoplastic (0.802) Apoptosis agonist (0.789)	Anti-inflammatory (0.786) Prostate disorders treatment (0.685)
**26**	Acute neurologic disorders treatment (0.867) Anti-neoplastic (0.812)	Diuretic (0.813) Male reproductive dysfunction treatment (0.759)
**27**	Anti-hypercholesterolemic (0.959)	Anti-neoplastic (0.832)

* Only activities with Pa > 0.7 are shown. The main biological activity has a value where Pa is more than 0.7.

**Table 2 molecules-28-05549-t002:** Biological activities of mono-aromatic steroids (**28–52**).

No.	Dominated Biological Activity (Pa) *	Additional Predicted Activities (Pa) *
**28**	Anti-hypercholesterolemic (0.961) Proliferative diseases treatment (0.711)	Anti-neoplastic (0.840) Apoptosis agonist (0.787)
**29**	Neuroprotector (0.979) Respiratory analeptic (0.970) Anti-neoplastic (0.888)	Anti-hypercholesterolemic (0.953) Anti-infective (0.933) Anti-protozoal (*Leishmania*) (0.922)
**30**	Anti-hypercholesterolemic (0.860) Anti-inflammatory (0.754)	Anti-neoplastic (0.805) Chemopreventive (0.721)
**31**	Anti-hypercholesterolemic (0.907) Anti-inflammatory (0.765)	Anti-neoplastic (0.836) Apoptosis agonist (0.788)
**32**	Anti-hypercholesterolemic (0.764)	Anti-inflammatory (0.695)
**33**	Anti-hypercholesterolemic (0.929)	Respiratory analeptic (0.885)
**34**	Anti-hypercholesterolemic (0.935)	Apoptosis agonist (0.850)
**35**	Anti-hypercholesterolemic (0.950) Anti-Parkinsonian, rigidity relieving (0.875)	Apoptosis agonist (0.898) Anti-neoplastic (0.880)
**36**	Anti-hypercholesterolemic (0.806)	Anti-neoplastic (0.729)
**37**	Anti-hypercholesterolemic (0.914) Hypolipemic (0.858)	Apoptosis agonist (0.894) Anti-neoplastic (0.879)
**38**	Anti-neoplastic (0.922)	Immunosuppressant (0.774)
**39**	Anti-neoplastic (0.899) Apoptosis agonist (0.896)	Anti-inflammatory (0.795)
**40**	Neuroprotector (0.829)	Anti-allergic (0.731)
**41**	Anti-convulsant (0.877)	
**42**	Apoptosis agonist (0.828) Anti-neoplastic (0.798)	Anti-inflammatory (0.813)
**43**	Anti-neoplastic (0.782)	Anti-bacterial (0.736)
**44**	Acute neurologic disorders treatment (0.867)	Anti-neoplastic (0.797)
**45**	Anti-inflammatory (0.825)	Apoptosis agonist (0.793)
**46**	Anti-neoplastic (0.884)	Apoptosis agonist (0.848)
**47**	Anti-neoplastic (0.799)	Apoptosis agonist (0.716)
**48**	Anti-neoplastic (0.858)	Anti-hypercholesterolemic (0.839)
**49**	Anti-neoplastic (0.858)	Cell adhesion molecule inhibitor (0.795)
**50**	Anti-neoplastic (0.841)	Immunosuppressant (0.722)
**51**	Anti-neoplastic (0.844)	Apoptosis agonist (0.792)
**52**	Apoptosis agonist (0.706)	Acute neurologic disorders treatment (0.768)

* Only activities with Pa > 0.7 are shown.

**Table 3 molecules-28-05549-t003:** Biological activities of aromatic steroids (**53**–**83**).

No.	Dominated Biological Activities (Pa) *	Additional Predicted Activities (Pa) *
**53**	Ovulation inhibitor (0.866)	Anti-neoplastic (0.824)
**54**	Acute neurologic disorders treatment (0.925) Anti-neoplastic (0.790)	Diuretic (0.824) Male reproductive dysfunction treatment (0.791)
**55**	Acute neurologic disorders treatment (0.826) Anti-neoplastic (0.818)	Respiratory analeptic (0.811) Neuroprotector (0.807)
**56**	Ovulation inhibitor (0.846); male reproductive dysfunction treatment (0.815)	Anti-neoplastic (0.821)
**57**	Neuroprotector (0.837) Anti-neoplastic (0.833)	Acute neurologic disorders treatment (0.828)
**58**	Ovulation inhibitor (0.843) Lipid metabolism regulator (0.723)	Anti-neoplastic (0.839) Neuroprotector (0.829)
**59**	Acute neurologic disorders treatment (0.932) Anti-neoplastic (0.810)	Laxative (0.833) Diuretic (0.751)
**60**	Apoptosis agonist (0.924) Anti-neoplastic (0.868)	Antioxidant (0.776) Neuroprotector (0.728)
**61**	Anti-osteoporotic (0.837)	Anti-neoplastic (0.735)
**62**	Anti-hypercholesterolemic (0.860)	Respiratory analeptic (0.847)
**63**	Anti-osteoporotic (0.776)	Anti-neoplastic (0.732)
**64**	Apoptosis agonist (0.758) Anti-neoplastic (0.733)	Anti-inflammatory (0.744)
**65**	Apoptosis agonist (0.758) Anti-neoplastic (0.733)	Anti-inflammatory (0.744)
**66**	Anti-inflammatory (0.807)	Apoptosis agonist (0.746); anti-neoplastic (0.726)
**67**	Anti-infertility, female (0.796)	Anti-inflammatory (0.794)
**68**	Anti-neoplastic (0.697)	Ovulation inhibitor (0.683)
**69**	Prostate disorders treatment (0.699)	Anti-inflammatory (0.661)
**70**	Anti-neoplastic (0.825) Alzheimer’s disease treatment (0.824)	Neurodegenerative diseases treatment (0.809) Psychotropic (0.700)
**71**	Anti-eczematic (0.767)	Anti-dyskinetic (0.670)
**72**	Anti-eczematic (0.695)	Autoimmune disorders treatment (0.652)
**73**	Anti-eczematic (0.767)	Anti-dyskinetic (0.670)
**74**	Anti-eczematic (0.782) Anti-psoriatic (0.619)	Anti-neurotic (0.709)
**75**	Neuroprotector (0.685)	Acute neurologic disorders treatment (0.647)
**76**	Hypolipemic (0.724)	Anti-convulsant (0.649)
**77**	Anti-eczematic (0.885) Anti-psoriatic (0.757)	Anti-inflammatory (0.735)
**78**	Anti-eczematic (0.709) Anti-psoriatic (0.632)	Anti-convulsant (0.661)
**79**	Anti-eczematic (0.691) Anti-psoriatic (0.622)	Psychotropic (0.611) Anti-convulsant (0.570)
**80**	Apoptosis agonist (0.758) Anti-neoplastic (0.733)	Anti-inflammatory (0.744)
**81**	Acute neurologic disorders treatment (0.778)	Neuroprotector (0.733)
**82**	Anti-inflammatory (0.650)	Menopausal disorders treatment (0.628)
**83**	Anti-inflammatory (0.782)	Anti-eczematic (0.771)

* Only activities with Pa > 0.7 are shown.

**Table 4 molecules-28-05549-t004:** Biological activities of steroid phosphate esters (**84**–**98**).

No.	Dominated Biological Activity (Pa) *	Additional Predicted Activities (Pa) *
**84**	Wound-healing agent (0.975) Hepatoprotectant (0.961) Analeptic (0.952) Laxative (0.933)	Anti-hypercholesterolemic (0.926) Anti-carcinogenic (0.912) Hemostatic (0.853) Anti-neoplastic (0.841)
**85**	Hepatoprotectant (0.874) Analeptic (0.874)	Anti-carcinogenic (0.861) Anti-neoplastic (0.848)
**86**	Wound-healing agent (0.947) Analeptic (0.941) Hepatoprotectant (0.932)	Anti-carcinogenic (0.915) Anti-hypercholesterolemic (0.912) Anti-neoplastic (0.843)
**87**	Anti-hypercholesterolemic (0.894) Hepatoprotectant (0.853) Wound-healing agent (0.844)	Anti-neoplastic (0.816) Anti-inflammatory (0.782) Cholesterol synthesis inhibitor (0.778)
**88**	Anti-hypercholesterolemic (0.894) Hepatoprotectant (0.853) Wound-healing agent (0.844)	Anti-neoplastic (0.816) Anti-inflammatory (0.782) Cholesterol synthesis inhibitor (0.778)
**89**	Anti-neoplastic (0.845) Anti-fungal (0.814)	Anti-inflammatory (0.693) Anti-bacterial (0.651)
**90**	Anti-fungal (0.837)	Anti-neoplastic (0.824)
**91**	Anti-neoplastic (0.827)	Anti-fungal (0.663)
**92**	Anti-neoplastic (0.852) Anti-neoplastic (liver cancer) (0.790)	Anti-eczematic (0.730) Anti-allergic (0.650)
**93**	Anti-neoplastic (0.852) Anti-neoplastic (liver cancer) (0.790)	Anti-eczematic (0.730) Anti-allergic (0.650)
**94**	Anti-neoplastic (0.827) Anti-neoplastic (liver cancer) (0.607)	Anti-fungal (0.663) Anti-bacterial (0.636)
**95**	Anti-neoplastic (0.841)	Anti-fungal (0.799)
**96**	Anti-fungal (0.850) Anti-bacterial (0.717)	Anti-neoplastic (0.832) Anti-carcinogenic (0.707)
**97**	Anti-fungal (0.850) Anti-bacterial (0.717)	Anti-neoplastic (0.832) Anti-carcinogenic (0.707)
**98**	Anti-fungal (0.858) Anti-bacterial (0.739)	Anti-neoplastic (0.842) Anti-carcinogenic (0.733)

* Only activities with Pa > 0.7 are shown.

**Table 5 molecules-28-05549-t005:** Biological activities of steroid phosphate esters (**99**–**114**).

No.	Dominated Biological Activity (Pa) *	Additional Predicted Activities (Pa) *
**99**	Anti-inflammatory (0.910) Anesthetic general (0.908)	Respiratory analeptic (0.904) Anti-osteoporotic (0.878)
**100**	Neuroprotector (0.987) Anesthetic general (0.959)	Respiratory analeptic (0.944) Anti-hypercholesterolemic (0.909)
**101**	Anesthetic general (0.991) Neuroprotector (0.976) Anti-inflammatory (0.906)	Respiratory analeptic (0.990) Anti-hypercholesterolemic (0.894)
**102**	Respiratory analeptic (0.979) Anesthetic general (0.973) Neuroprotector (0.972)	Anti-hypercholesterolemic (0.971) Wound-healing agent (0.913) Anti-neoplastic (0.826)
**103**	Respiratory analeptic (0.995) Anesthetic general (0.948) Wound-healing agent (0.897)	Anti-hypercholesterolemic (0.945) Neuroprotector (0.932) Hemostatic (0.910)
**104**	Respiratory analeptic (0.995) Anti-hypercholesterolemic (0.967) Anesthetic general (0.954)	Hemostatic (0.928) Wound-healing agent (0.921) Neuroprotector (0.909)
**105**	Anti-hypercholesterolemic (0.996) Cholesterol absorption inhibitor (0.976) Cholesterol synthesis inhibitor (0.952) Lipid metabolism regulator (0.952)	Acute neurologic disorders treatment (0.948) Anti-hyperlipoproteinemic (0.920) Hypolipemic (0.919) Respiratory analeptic (0.908)
**106**	Anti-hypercholesterolemic (0.999) Anti-hyperlipoproteinemic (0.986) Hypolipemic (0.974)	Cholesterol absorption inhibitor (0.957) Lipid metabolism regulator (0.954) Cholesterol synthesis inhibitor (0.916)
**107**	Anti-neoplastic (0.822)	Anti-inflammatory (0.645)
**108**	Neuroprotector (0.982) Anesthetic general (0.931)	Anti-hypercholesterolemic (0.909)
**109**	Anesthetic general (0.970) Neuroprotector (0.965)	Respiratory analeptic (0.961) Acute neurologic disorders treatment (0.916)
**110**	Anti-inflammatory (0.979) Anti-allergic (0.959)	Anti-asthmatic (0.951) Anti-arthritic (0.944)
**111**	Respiratory analeptic (0.929) Anti-ischemic, cerebral (0.907)	Anesthetic general (0.897) Anti-neoplastic (0.847)
**112**	Anti-ischemic, cerebral (0.979) Respiratory analeptic (0.919)	Anti-osteoporotic (0.843) Anesthetic general (0.830)
**113**	Respiratory analeptic (0.937) Anti-ischemic, cerebral (0.922)	Anesthetic general (0.897)
**114**	Anti-ischemic, cerebral (0.978) Respiratory analeptic (0.911)	Anti-osteoporotic (0.852)

* Only activities with Pa > 0.7 are shown.

**Table 6 molecules-28-05549-t006:** Biological activities of chlorinated plant steroids (**115**–**133**).

No.	Dominated Biological Activity (Pa) *	Additional Predicted Activities (Pa) *
**115**	Hepatic disorders treatment (0.940) Anti-eczematic (0.924)	Macular degeneration treatment (0.921) Cytostatic (0.904)
**116**	Hepatic disorders treatment (0.933) Anti-eczematic (0.932)	Macular degeneration treatment (0.926) Cytostatic (0.875)
**117**	Anti-eczematic (0.919) Hepatic disorders treatment (0.908)	Cytostatic (0.921) Macular degeneration treatment (0.912)
**118**	Anti-diabetic (0.938) Myocardial infarction treatment (0.823)	Anti-eczematic (0.902) Alzheimer’s disease treatment (0.664)
**119**	Anti-diabetic (0.981) Lipoprotein disorders treatment (0.938)	Anti-eczematic (0.902) Alzheimer’s disease treatment (0.666)
**120**	Anti-diabetic (0.980) Lipoprotein disorders treatment (0.939)	Anti-eczematic (0.897) Alzheimer’s disease treatment (0.696)
**121**	Apoptosis agonist (0.888) Anti-neoplastic (0.860)	Anti-eczematic (0.910) Cytostatic (0.643)
**122**	Neurodegenerative diseases treatment (0.913) Alzheimer’s disease treatment (0.889)	Anti-eczematic (0.926) Anti-Parkinsonian (0.856)
**123**	Lipoprotein disorders treatment (0.968) Anti-diabetic (0.953)	Anti-eczematic (0.912) Alzheimer’s disease treatment (0.670)
**124**	Anti-eczematic (0.930) Myocardial infarction treatment (0.872)	Anti-neoplastic (0.866) Cytostatic (0.819)
**125**	Anti-eczematic (0.823) Allergic conjunctivitis treatment (0.629)	Anti-neoplastic (0.785) Anti-inflammatory (0.731)
**126**	Myocardial infarction treatment (0.825) Anti-neoplastic (0.707)	Anti-eczematic (0.815) Allergic conjunctivitis treatment (0.618)
**127**	Anti-neoplastic (0.918) Apoptosis agonist (0.793) Anti-neoplastic (myeloid leukemia) (0.520)	Respiratory analeptic (0.757) Anti-secretoric (0.755) Lipid metabolism regulator (0.677)
**128**	Anti-neoplastic (0.892) Apoptosis agonist (0.796) Anti-metastatic (0.551)	Hepatoprotectant (0.739) Hepatic disorders treatment (0.701) Dermatologic (0.614)
**129**	Cytostatic (0.863) Anti-neoplastic (0.826) Apoptosis agonist (0.797)	Anti-eczematic (0.929) Macular degeneration treatment (0.856) Alzheimer’s disease treatment (0.729)
**130**	Lipoprotein disorders treatment (0.952) Anti-diabetic (0.943) Anti-asthmatic (0.593)	Anti-eczematic (0.904) Anti-neoplastic (0.765) Anti-leukemic (0.651)
**131**	Insulin promoter (0.986) Myocardial infarction treatment (0.868) Anti-neoplastic (0.833) Apoptosis agonist (0.768)	Anti-eczematic (0.910) Anti-fungal (0.670) Anti-psoriatic (0.582) Anti-bacterial (0.535)
**132**	Anti-eczematic (0.914) Anti-fungal (0.795) Anti-parasitic (0.756)	Anti-neoplastic (0.854) Apoptosis agonist (0.786) Cytostatic (0.722)
**133**	Anti-neoplastic (0.914) Apoptosis agonist (0.823)	Anti-asthmatic (0.834) Anti-allergic (0.828)

* Only activities with Pa > 0.7 are shown.

**Table 7 molecules-28-05549-t007:** Biological activities of chlorinated plant steroids (**134–153**).

No.	Dominated Biological Activity (Pa) *	Additional Predicted Activities (Pa) *
**134**	Apoptosis agonist (0.806) Anti-neoplastic (0.803)	Genital warts treatment (0.724) Anti-eczematic (0.718)
**135**	Insulin promoter (0.981) Myocardial infarction treatment (0.819)	Anti-neoplastic (0.797) Apoptosis agonist (0.695)
**136**	Insulin promoter (0.986) Myocardial infarction treatment (0.899)	Anti-neoplastic (0.866) Apoptosis agonist (0.772)
**137**	Insulin promoter (0.986) Myocardial infarction treatment (0.899)	Anti-neoplastic (0.839) Apoptosis agonist (0.696)
**138**	Anti-neoplastic (0.875) Apoptosis agonist (0.795)	Anti-asthmatic (0.816) Anti-allergic (0.533)
**139**	Anti-neoplastic (0.885) Apoptosis agonist (0.824)	Anti-psoriatic (0.595) Anti-allergic (0.539)
**140**	Anti-neoplastic (0.806) Apoptosis agonist (0.634)	Myocardial infarction treatment (0.781) Hypolipemic (0.599)
**141**	Hepatic disorders treatment (0.934) Immunosuppressant (0.691)	Anti-allergic (0.618) Allergic conjunctivitis treatment (0.543)
**142**	Hepatic disorders treatment (0.942) Anti-neoplastic (0.782)	Anti-allergic (0.758) Anti-asthmatic (0.728)
**143**	Hepatic disorders treatment (0.930) Anti-neoplastic (0.753)	Anti-allergic (0.711) Allergic conjunctivitis treatment (0.597)
**144**	Anti-neoplastic (0.888) Apoptosis agonist (0.761)	Anti-inflammatory (0.815) Anti-fungal (0.629)
**145**	Anti-neoplastic (0.907) Apoptosis agonist (0.673)	Anti-inflammatory (0.824) Anti-fungal (0.597)
**146**	Anti-eczematic (0.850) Anti-neoplastic (0.765)	Allergic conjunctivitis treatment (0.649) Anti-allergic (0.641)
**147**	Anti-eczematic (0.850) Anti-pruritic (0.787)	Allergic conjunctivitis treatment (0.649) Anti-allergic (0.641)
**148**	Anti-protozoal (0.956) Genital warts treatment (0.824)	Anti-neoplastic (0.761) Anti-metastatic (0.530)
**149**	Anti-protozoal (0.954) Genital warts treatment (0.805)	Anti-neoplastic (0.759) Apoptosis agonist (0.540)
**150**	Anti-protozoal (0.958) Anti-protozoal (*Plasmodium*) (0.953)	Genital warts treatment (0.798) Anti-neoplastic (0.766)
**151**	Insulin promoter (0.984) Cytostatic (0.907)	Anti-eczematic (0.907) Anti-fungal (0.752)
**152**	Insulin promoter (0.982) Cytostatic (0.921)	Anti-eczematic (0.919) Macular degeneration treatment (0.912)
**153**	Anti-eczematic (0.922) Macular degeneration treatment (0.913)	Anti-neoplastic (0.868) Cytostatic (0.866)

* Only activities with Pa > 0.7 are shown.

**Table 8 molecules-28-05549-t008:** Biological activities of halogenated steroids (**154–175**).

No.	Dominated Biological Activity (Pa) *	Additional Predicted Activities (Pa) *
**154**	Anti-neoplastic (0.860) Prostate disorders treatment (0.781)	Bone diseases treatment (0.722) Anti-inflammatory (0.639)
**155**	Anti-neoplastic (0.894) Prostate disorders treatment (0.799)	Bone diseases treatment (0.787) Anti-inflammatory (0.731)
**156**	Anti-neoplastic (0.934) Prostate cancer treatment (0.885) Anti-neoplastic (sarcoma) (0.875) Anti-neoplastic (renal cancer) (0.820)	Choleretic (0.879) Anti-hypercholesterolemic (0.828) Anti-fungal (0.781) Dermatologic (0.778)
**157**	Anti-neoplastic (0.922) Anti-neoplastic (sarcoma) (0.836)	Anti-osteoporotic (0.803) Bone diseases treatment (0.781)
**158**	Anti-hypercholesterolemic (0.937) Atherosclerosis treatment (0.831)	Respiratory analeptic (0.878) Anti-infertility, female (0.833)
**159**	Anti-neoplastic (0.881) Growth stimulant (0.751)	Dermatologic (0.771) Anti-fungal (0.696)
**160**	Anti-hypercholesterolemic (0.885)	Anesthetic general (0.823)
**161**	Anti-neoplastic (0.810) Apoptosis agonist (0.776)	Prostate disorders treatment (0.688) Acute neurologic disorders treatment (0.680)
**162**	Anti-neoplastic (0.805) Apoptosis agonist (0.744) Cytoprotectant (0.690) Prostate disorders treatment (0.681)	Dermatologic (0.750) Anti-viral (influenza) (0.738) Anti-bacterial (0.736) Anti-fungal (0.728)
**163**	Anti-neoplastic (0.805) Apoptosis agonist (0.744) Cytoprotectant (0.690) Prostate disorders treatment (0.681)	Dermatologic (0.750) Anti-viral (influenza) (0.738) Anti-bacterial (0.736) Anti-fungal (0.728)
**164**	Anti-neoplastic (0.851) Anti-carcinogenic (0.754)	Biliary tract disorders treatment (0.841) Bone diseases treatment (0.725)
**165**	Anti-neoplastic (0.882) Cytostatic (0.793)	Anti-bacterial (0.736) Anti-fungal (0.695)
**166**	Anti-neoplastic (0.822) Cytostatic (0.782)	Anti-parasitic (0.718) Anti-protozoal (0.714)
**167**	Glucan endo-1,3-b-D-glucosidase inhibitor (0.890)	Biliary tract disorders treatment (0.845)
**168**	Anti-neoplastic (0.884)	Anti-inflammatory (0.829)
**169**	Anti-inflammatory (0.829) Anti-viral (0.826)	Anti-neoplastic (0.784) Apoptosis agonist (0.763)
**170**	Anti-hypercholesterolemic (0.941) Atherosclerosis treatment (0.831)	Anti-infertility, female (0.833) Prostate disorders treatment (0.773)
**171**	Anti-neoplastic (0.912) Cytoprotectant (0.764) Prostate disorders treatment (0.767)	Respiratory analeptic (0.894) Erythropoiesis stimulant (0.776)
**172**	Anti-neoplastic (0.912) Cytoprotectant (0.764) Prostate disorders treatment (0.767)	Respiratory analeptic (0.894) Erythropoiesis stimulant (0.776) Apoptosis agonist (0.677)
**173**	Anti-hypercholesterolemic (0.911) Myocardial infarction treatment (0.900) Atherosclerosis treatment (0.811)	Apoptosis agonist (0.862) Anti-neoplastic (0.846) Prostate disorders treatment (0.823)
**174**	Respiratory analeptic (0.911) Myocardial infarction treatment (0.906)	Anti-hypercholesterolemic (0.845) Anti-diabetic (type 2) (0.669)
**175**	Myocardial infarction treatment (0.864) Immunosuppressant (0.734)	Dermatologic (0.785) Anti-psoriatic (0.728)

* Only activities with Pa > 0.7 are shown.

## Data Availability

Not applicable.

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
