# Peer review of "Biological Activity and Structural Diversity of Steroids Containing Aromatic Rings, Phosphate Groups, or Halogen Atoms"

_molecules, 2023, doi:10.3390/molecules28145549_

Round 1

Reviewer 1 Report

The manuscript is a review of bioactive steroids from natural sources. The review is quite comprehensive, but at times hard to follow. It is especially not clear how the Pa values in the various tables were calculated, or if these are from the literature.

In line 48 I would disagree with the definition of phytosterols, they do not need to have bioactivity. For instance sigmasterol is a membrane component in a large number of plants and therefore a phytosterol, but I would not say that it has bioactivity in the traditional sense.

Author Response

Reviewer 1

The author is grateful to Reviewer 1 for reading the manuscript and comments. Pa values in various tables and whether they are taken from the literature. Since this is a review, we provide links to publications.

Regarding phytosterols. Yes, you are correct that not all phytosterols need to be active.

Reviewer 2 Report

The review entitled “Bioactive Steroids derived from Terrestrial and Marine Environments” is described in detail about the structural diversity and biological activity of various natural steroids of isoprenoid lipids, halogenated, phosphate esters etc. derived from plants, algae and marine invertebrates.

The figures in page 5 (line 103) and page 6 (line 116)  has same figure number: Fig. 3. Please correct it  and change the remaining figure numbers.

Fig. 11 and 23 Structures looks like stretched horizontally. Please redraw the structures correctly as in the Fig. 3 & 10

The beauty in the review is hidden in the colorful pictures, images and structures. It can easily attracts the readers and easily understandable. The author did excellent work in drawing typical diagrams and collecting high resolution images. But it would be more appropriate to maintain the same integrity throughout the manuscript. Hence, I recommend the author to redraw the structures in Fig. 11, 14, 18, 23, 27, 31. Please redraw the structures as in the Fig. 3 and 10 and maintain the same structure size.

It would be more appropriate to include description of the figures in the discussion part instead of writing the lengthy figure captions for the figures in Fig. 2, Fig. 3 (in page 6), Fig. 4, 5, 6, 7, 8, 13, 15. 16, 19, 20, 21, 24, 25, 26, 28, 29, 30, 32, 33, 34.

Other than these corrections, the looks very good and well explained.

Author Response

Reviewer 2

The author is grateful to Reviewer 2 for reading the manuscript and comments. We have edited and highlighted in blue.

Figures the structures in Fig. 11, 14, 18, 23, 27, 31 corrected. To do this, you do not need to redraw them, they have the option to change in different directions. Marked in blue.

I could agree with the reviewer that it would be more appropriate to include a description of the figures in the discussion section instead of writing long figure captions. However, we believe that the description given under the figures gives a complete understanding of the material presented. Therefore, we left signatures under figures 4, 5, 6, 7, 8, 13, 15. 16, 19, 20, 21, 24, 25, 26, 28, 29, 30, 32, 33, 34.
